# On Good Practices for Task-Specific Distillation of Large Pretrained Visual Models

**Juliette Marrie**                                              *juliette.marrie@naverlabs.com*
*NAVER LABS Europe*
*Univ. Grenoble Alpes, Inria, CNRS, Grenoble INP, LJK, 38000 Grenoble*

**Michael Arbel**                                                   *michael.arbel@inria.fr*
*Univ. Grenoble Alpes, Inria, CNRS, Grenoble INP, LJK, 38000 Grenoble*

**Julien Mairal**                                                   *julien.mairal@inria.fr*
*Univ. Grenoble Alpes, Inria, CNRS, Grenoble INP, LJK, 38000 Grenoble*

**Diane Larlus**                                                 *diane.larlus@naverlabs.com*
*NAVER LABS Europe*

**Reviewed on OpenReview:** *https://openreview.net/forum?id=oyISaaeHwD*

## Abstract

Large pretrained visual models exhibit remarkable generalization across diverse recognition tasks. Yet, real-world applications often demand compact models tailored to specific problems. Variants of knowledge distillation have been devised for such a purpose, enabling task-specific compact models (the students) to learn from a generic large pretrained one (the teacher). In this paper, we show that the excellent robustness and versatility of recent pretrained models challenge common practices established in the literature, calling for a new set of optimal guidelines for task-specific distillation. To address the lack of samples in downstream tasks, we also show that a variant of Mixup based on stable diffusion complements standard data augmentation. This strategy eliminates the need for engineered text prompts and improves distillation of generic models into streamlined specialized networks.[1]

## 1 Introduction

Recent large pretrained visual models demonstrate robust generalization across diverse computer vision tasks. Developed by leveraging substantial computational resources, these models are trained on enormous (often internal) sets of visual data, enabling them to learn rich visual representations. Such models exhibit remarkable transfer performance on downstream tasks with frozen features, achieving competitive results through simple linear probing (see, *e.g.* Li et al., 2022; He et al., 2022; Fang et al., 2023; Oquab et al., 2024). However, the size of the best performing models often poses limitations for various real-world applications, both in terms of inference time and memory usage, especially in scenarios with constrained resources.

An essential question thus emerges: *How can we most effectively transfer the rich visual representations from these large models to a smaller architecture?* While smaller models distilled from the larger ones on a sizeable generic dataset are sometimes available (Oquab et al., 2024), is simply finetuning them to specific tasks optimal? As visual pretrained models are becoming larger, the cost of finetuning them is often out of reach for many users. It is therefore natural to ask whether a teacher trained with simple probing (linear or with a small multilayer perceptron) is sufficiently competent to guide the training of a smaller model, specialized for a given computer vision task. Finally, as distillation often benefits from data augmentation (Beyer et al., 2022), and given the effectiveness of data augmentation methods based on Stable Diffusion (Rombach et al.,

---

[1]Project page: `https://europe.naverlabs.com/tskd`

2022; Saharia et al., 2022b) in supervised learning (Trabucco et al., 2023; Azizi et al., 2023; Zhou et al., 2022), leveraging generative models for distillation seems promising. However, unlike in supervised learning where the generative model is usually conditioned by text prompts, *e.g.*, with class labels, the dependence on class information becomes questionable when used for distillation, as labels are not technically required. This raises the question of how to best leverage these models in the context of knowledge distillation.

In this paper, we study these fundamental questions. (i) We delineate optimal practices for leveraging large pretrained visual models in real-world applications constrained by limited resources, supported by an extensive experimental analysis. Our work shows that a simple, cost-efficient approach to supervised distillation from large pretrained models consistently achieves superior results. (ii) We investigate various data augmentation strategies based on Stable Diffusion and demonstrate that a variation of Mixup is notably efficient for distillation. Originally proposed by Pinkney (2022) in a different context to generate visually appealing combinations of images, it proves particularly effective when employed as a data augmentation technique for distillation. It operates solely on unlabeled images, eliminating the necessity for text prompt engineering, and remains agnostic to the downstream task.

Concretely, our work reaches a series of experimental conclusions that ground our guidelines. Our experiments are conducted using DINOv2 teachers (Oquab et al., 2024), recognized for providing strong baselines (see Section 4.2) and extended to EVA-02 MIM- and CLIP-pretrained models (Fang et al., 2023; Sun et al., 2023) (see Appendix B.3). Our findings, summarized below, are validated across different architectures and various tasks: classification on specific image modalities, fine-grained classification, and semantic segmentation.

1. *Probing can yield better teachers than finetuning.* The remarkable adaptability of recent large-scale pretrained models, such as DINOv2, challenges the need for finetuning the teacher, which is standard practice in prior research on task-specific distillation (Jiao et al., 2020; Sun et al., 2019; Touvron et al., 2021; Beyer et al., 2022; Huang et al., 2023).

2. *Task-specific distillation complements task-agnostic distillation.* Task-specific distillation allows transferring task-specific knowledge, leading to better representations compared to simply finetuning the student after task-agnostic distillation, as illustrated in Figure 1. We show that task-specific distillation consistently outperforms simple finetuning, which aligns with conclusions from prior works (Jiao et al., 2020; Huang et al., 2023), drawn for teachers finetuned on the target task. Our study extends their results to teachers that are only probed for the task, thus reducing the cost of the distillation procedure.

3. *Teachers do not need to be as accurate as their students.* This observation generalizes conclusions from early works (Yuan et al., 2020; Furlanello et al., 2018), conducted with teacher/student CNN models trained from scratch in a supervised manner. We show that even when DINOv2's pretrained ViT-S outperforms its teacher with simple finetuning, distillation can still be beneficial.

4. *Small models can directly learn from much larger ones.* Prior works suggest that a large capacity gap between teacher and student hinders distillation, and employ a middle-sized 'teacher assistant' to learn from the large model and teach the small one (Jiao et al., 2020; Mirzadeh et al., 2020; Wang et al., 2020). However, DINOv2's ViT-S was directly distilled from their ViT-g and yet demonstrates excellent generalization capabilities. Similarly, we show that task-specific distillation works equally well when using DINOv2's ViT-g or their middle-sized ViT-L to teach ViT-S.

5. *Diffusion models can be effectively leveraged as data augmentation for distillation without relying on class information*, making them applicable to tasks where text-conditioned image generation is non-trivial (such as semantic segmentation). To bypass the need for class information in Stable Diffusion, we leverage a diffusion model that generates *mixed* images conditionally on multiple images provided as input, taking inspiration from the classical Mixup augmentation. We show that, while being ineffective in the context of supervised learning, this mixing strategy consistently helps task-specific distillation.

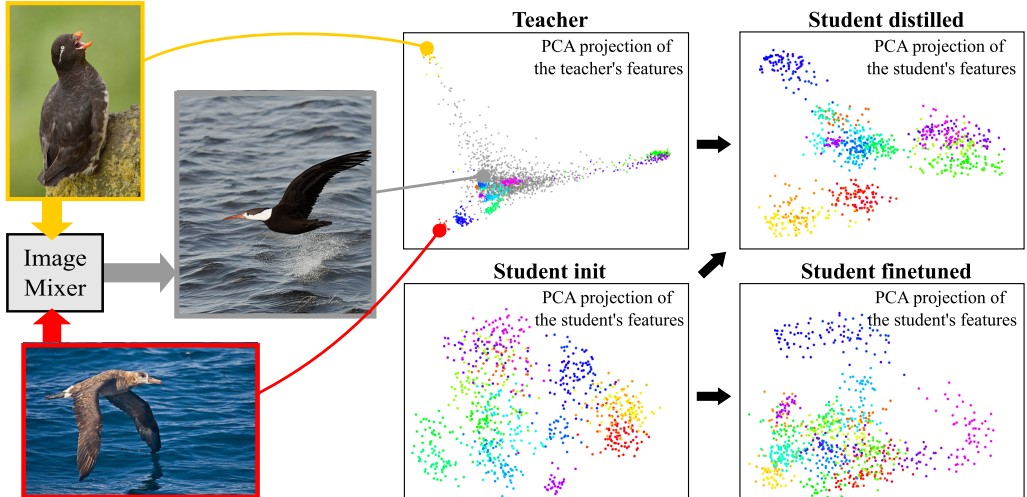

Figure 1: This paper advocates for distilling a large pretrained teacher (top, left) to train a small task-specific student model (top, right). This distillation process results in a better clustering of the representations compared to simply finetuning the student on the task (bottom, right). Distillation is improved by a class-agnostic data augmentation based on Stable Diffusion that consists in mixing real images to create synthetic ones, producing features shown in gray in the teacher plot. Each plot shows image features for 30 classes of the CUB Bird dataset, after PCA (one color per class).

## 2 Related work

In this section, we first discuss relevant prior work on knowledge distillation (Section 2.1). We then cover works that leverage Stable Diffusion for data augmentation, and discuss data augmentation in the context of distillation (Section 2.2).

### 2.1 Knowledge distillation

**Task-specific vs. generic distillation.** Following the pioneering work of Hinton et al. (2015), distillation has become a standard approach to transfer knowledge from one model into another (see Gou et al. 2021; Wang & Yoon 2021 for detailed surveys). Initially, knowledge distillation was conceived as a method to transfer knowledge from a large teacher network trained on a specific task to a small student network (Hinton et al., 2015; Ba & Caruana, 2014). With the rise of self-supervised learning, the approach was extended to transfer general representations produced by a large generic model into small ones (Abbasi Koohpayegani et al., 2020; Fang et al., 2021; Xu et al., 2022; Gao et al., 2022; Navaneet et al., 2021; Wu et al., 2022; Duval et al., 2023). There, distillation is used as a knowledge compression mechanism, which is motivated by the observation that directly pretraining small models on large amounts of data leads to underwhelming results compared to learning them by distillation from large pretrained models (Abbasi Koohpayegani et al., 2020; Fang et al., 2021; Xu et al., 2022; Wu et al., 2022; Oquab et al., 2024).

In the context of self-supervised learning, it is then common to finetune distilled models on various downstream tasks, without further exploiting the teacher's knowledge (Sun et al., 2019; Touvron et al., 2021; Beyer et al., 2022). Surprisingly, only few studies have explored a task-specific distillation procedure that leverages both the teacher and the downstream task. An example is the two-stage distillation introduced in natural language processing by Jiao et al. (2020) and recently applied to vision tasks by Huang et al. (2023). Specifically, their approach involves a conventional generic distillation, followed by finetuning the teacher on a downstream task and applying a second task-specific distillation involving the finetuned teacher. In contrast, our findings indicate that finetuning the teacher is not always the optimal strategy, and we advocate for a less computationally demanding approach.

**Architecture-dependent distillation.** Some variants of knowledge distillation directly exploit the specific architecture of both the teacher and the student. These include feature-based knowledge distillation often tailored to CNNs (Romero et al., 2015; Zagoruyko & Komodakis, 2017; Chen et al., 2021a;b), where knowledge is distilled by matching representations from any intermediate layer(s), or aligning mutual relations in the feature space (Yim et al., 2017; Tung & Mori, 2019). Approaches specific to transformers have also emerged, consisting, for instance, of adding a separate distillation token (Touvron et al., 2021). Simultaneously, other works have proposed architecture-agnostic distillation approaches relying on particular loss functions (Tian et al., 2020; Zhao et al., 2022). For example, Tian et al. (2020) propose a contrastive objective inspired by self-supervised learning approaches. In our work, we adopt a task- and architecture-agnostic distillation framework, therefore bypassing the need for adjusting to the model's architecture.

## 2.2 Data augmentation

Traditionally, data augmentation has been used to improve the generalization capabilities of deep neural networks (Wang & Yoon, 2021). Recently, Stable Diffusion models have emerged as another compelling tool for data augmentation, and have been broadly studied in the context of supervised learning. In the context of knowledge distillation, data augmentation is not constrained by the need of class labels or segmentation masks, which suggests that optimal augmentation approaches may differ from those delineated in the context of supervised learning. Below we discuss prior works on the use of Stable Diffusion for data augmentation and prior studies on data augmentation for knowledge distillation.

**Data augmentation with Stable Diffusion.** Recent generative models such as latent diffusion models (Rombach et al., 2022) have emerged as a compelling way to artificially augment training data (Trabucco et al., 2023; Azizi et al., 2023; Dunlap et al., 2024) or even replace it (Sarıyıldız et al., 2023), usually using class names as textual prompts. Yet designing prompts can be difficult for tasks such as segmentation, as it requires featuring the multiple classes found in an image. Prior works often resort to prompt engineering (Fang et al., 2024) or to language models to generate prompts from class names (Nguyen et al., 2023; Zhou et al., 2022).

An alternative to text-to-image generation is to leverage image-to-image diffusion models to directly provide training images as prompts. Image-to-image diffusion models have proven successful at various tasks such as restoration (Saharia et al., 2022a) or image editing (Brooks et al., 2023). However, using them as a tool for data augmentation raises significant challenges. These models can struggle with producing meaningful variations such as viewpoint changes or object shape variations, as pointed out by Brooks et al. (2023). Properties such as object shape, location, and appearance can be extracted and controlled from the internal representations of diffusion models (Epstein et al., 2023) but this requires manual interventions and cannot be universally applied to any task. For dense segmentation tasks, Yang et al. (2023) propose to generate synthetic data based on the segmentation mask of real images. This approach allows generating image/mask pairs without resorting to prompt engineering, but is restricted to supervised tasks with access to segmentation masks, and synthetic images are bound to be generated with these fixed masks. In contrast, we advocate an approach that can be universally applied to any task and that produces substantial image variations by interpolating between multiple training images (Pinkney, 2022).

**Data augmentation in the context of distillation.** In the context of knowledge distillation, Beyer et al. (2022) recommend to apply the same augmentations to the inputs of both teacher and student networks to ensure they are provided with consistent views. Wang et al. (2022) suggest that a good data augmentation scheme should reduce the covariance of the teacher-student cross-entropy, and propose an enhanced CutMix augmentation. Alternatively, Stanton et al. (2021) show the positive impact of Mixup on knowledge distillation. In this work, we show that distillation works better when performing data augmentation that goes beyond simple photometric and geometric transformations such as vanilla Mixup or CutMix, by exploiting the richness of generative models such as Stable Diffusion.

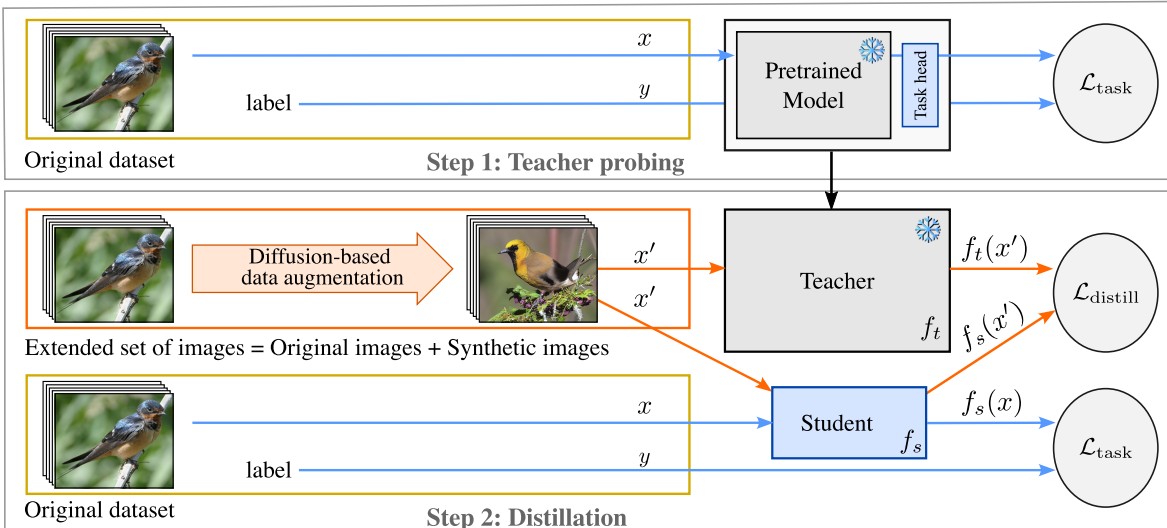

Figure 2: **Overview of the task-specific distillation pipeline.** The pretrained model is probed to build a teacher (top). Then its knowledge is distilled (Hinton et al., 2015) by minimizing the distillation loss $\mathcal{L}_{\text{distill}}$ jointly with the task loss $\mathcal{L}_{\text{task}}$ (bottom). $\mathcal{L}_{\text{distill}}$ is optimized with both i) original images $x$ and ii) synthetic images obtained with Stable Diffusion $x'$, while $\mathcal{L}_{\text{task}}$ is only optimized on the original dataset $(x, y)$. Note that $x$ and $x'$ are also transformed using standard data augmentation (not shown here).

## 3 Method

Our study focuses on task-specific distillation, which consists in training a small model for a specific supervised task while transferring knowledge from a large pretrained encoder. In Section 3.1, we detail the standard approach for task-specific distillation of pretrained models. It usually divides in two steps: first training a teacher model on the target task, then transferring the knowledge from the trained teacher to a student. Unlike prior work, our distillation is performed without teacher finetuning: we only train a *task head* on the pretrained encoder. Finetuning can be computationally expensive, especially when dealing with large teachers such as ViT-g, but it may also compromise the quality of the visual representation acquired during pretraining (*e.g.*, from self-supervision). In Section 3.2, we present a mixing data augmentation based on Stable Diffusion that leverages the teacher's knowledge more effectively to enhance the distillation process. The overall method is illustrated in Figure 2.

### 3.1 Task-specific distillation

We consider a task, such as classification or segmentation, where the goal is to predict a label $y$ (e.g. a class or a segmentation map) given an input image $x$. Typically, one could learn a model $f$ to perform such a task using a training set $\mathcal{D}_{\text{train}}$ of image/labels pairs $(x, y)$ by simply optimizing the following training loss (which is possibly regularized):

$$\mathcal{L}_{\text{task}}(f) = \mathbb{E}_{(x,y) \sim \mathcal{D}_{\text{train}}} \ell_{\text{task}}\left(f(x), y\right). \tag{1}$$

One can directly leverage a pretrained model by using it to initialize the model $f$, and then performing either *finetuning* or *probing* using objective $\mathcal{L}_{\text{task}}(f)$. However, these direct approaches require using the same architecture as the original pretrained model, which can be limiting for applications where inference speed and memory are critical factors. Instead, we are interested in learning a lightweight model $f_s$ that can still leverage knowledge from a much larger pretrained encoder model $e_t$ to perform the task. To this end, we first construct a teacher model $f_t$ from the pretrained encoder $e_t$ and then use it to distill knowledge relevant to the task on the lightweight model $f_s$.

**Step 1: Teacher probing.** We augment the encoder model $e_t$ with a task-specific prediction head $p_t$ creating the *teacher model* $f_t$ (i.e., $f_t(x) = p_t(e_t(x))$). The teacher is then *probed* for the supervised task by training the prediction head $p_t$ to minimize the training loss $\mathcal{L}_{\text{task}}(f_t)$. Notably, the parameters of the encoder $e_t$ remain frozen. This not only significantly reduces the training cost compared to finetuning but it also helps preserving information acquired during (self-supervised) pretraining. Our experiments (Table 2) indeed show that this probed teacher leads to better distillation results than its finetuned version in general.

**Step 2: Distillation.** After probing the teacher $f_t$, we use it to guide the training of a smaller *student model* $f_s$ on the downstream task. Specifically, we supplement the task loss $\mathcal{L}_{\text{task}}$ with a *distillation loss* $\mathcal{L}_{\text{distill}}$ that encourages the student's predictions to match the teachers', resulting in an overall objective of the form:

$$\mathcal{L}(f_s) := (1 - \alpha)\mathcal{L}_{\text{task}}(f_s) + \alpha\mathcal{L}_{\text{distill}}(f_s, f_t), \tag{2}$$

where $\alpha$ is a weighting parameter controlling the strength of the distillation loss. We define $\mathcal{L}_{\text{distill}}$ as an average of some dissimilarity measure $\ell_{\text{d}}$ between the student's and the teacher's predictions over a set $\mathcal{D}$ of *well-chosen* images:

$$\mathcal{L}_{\text{distill}}(f_s, f_t) = \mathbb{E}_{(x,.)\sim\mathcal{D}}\big[\ell_{\text{d}}\left(f_s(x), f_t(x)\right)\big]. \tag{3}$$

The loss in eq. (3) ensures our distillation protocol is agnostic to the architecture since the dissimilarity measure $\ell_{\text{d}}$ depends solely on the student's and the teacher's outputs and not on their internal structure. We set the dissimilarity measure $\ell_{\text{d}}$ to be the KL-divergence rescaled by a temperature parameter $T$ as proposed by Hinton et al. (2015):

$$\ell_{\text{d}}(f_s(x), f_t(x)) = T^2 D_{\text{KL}}\left(\frac{f_s(x)}{T}\,\bigg|\bigg|\,\frac{f_t(x)}{T}\right). \tag{4}$$

The choice of images $\mathcal{D}$ in the distillation loss (eq. (3)) is crucial as it defines the nature of images for which the student is required to match the teacher's predictions. While it is only natural to define $\mathcal{D}$ as the set of training images $\mathcal{D}_{\text{train}}$, this choice is not necessarily the most effective for extracting relevant knowledge from the teacher as $\mathcal{D}_{\text{train}}$ could offer a view that is too *narrow*. Instead, we propose to build $\mathcal{D}$ by extending $\mathcal{D}_{\text{train}}$ using an augmentation protocol based on Stable Diffusion, described in the next subsection.

## 3.2 Distilling with synthetic data

The distillation process outlined in Section 3.1 aims to align the teacher's and student's outputs for a set of images $\mathcal{D}$ that is sufficiently large and diverse to extract relevant knowledge. While it is possible to augment a dataset with standard data augmentation, our experiments indicate that this may not introduce enough diversity. When aiming for increased diversity, generating images relevant to the task—with suitable semantics and originating from the correct domain—is crucial. However, this step should be task-agnostic to avoid the need for manual tailoring to each downstream task, or to avoid providing class names or any other ground truth.

We propose to use a variant of Stable Diffusion, originally introduced by Pinkney (2022) for aesthetic purposes, named ImageMixer. It is a finetuned version of Rombach et al. (2022) that enables the *mixing* of CLIP image representations from two or more input images to generate a new one. More precisely, CLIP embeddings are concatenated along the sequence dimension and serve as a conditional input. We use this method as a variant of Mixup for data augmentation, which involves mixing random pairs of images, regardless of their classes. This enables us to create an augmented dataset $\mathcal{D}_{\text{sd}}$ containing both the original images from $\mathcal{D}_{\text{train}}$ and the synthetic ones. During training, we use $\mathcal{D}_{\text{sd}}$ by randomly sampling synthetic images and original ones with equal frequency.

Example images generated for the CUB (Wah et al., 2011), Pascal VOC (Everingham et al., 2010) and DomainNet's Painting (Peng et al., 2019) datasets can be found in Figure 3. Additional examples can be found in the appendix.

It is crucial to note that the corresponding augmented set is exclusively used for the distillation loss $\mathcal{L}_{\text{distill}}$. We have experimentally observed that introducing synthetic data in the optimization of $\mathcal{L}_{\text{task}}$ degrades

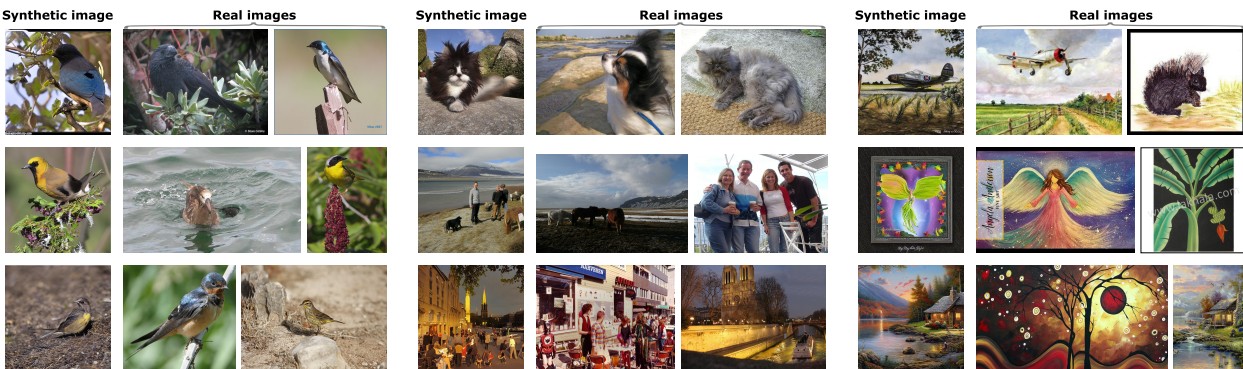

Figure 3: **Diffusion-based data augmentation**. Examples of synthetic images generated using ImageMixer (Pinkney, 2022) as described in Section 3.2, mixing two training images from CUB (Wah et al., 2011) (left), Pascal VOC (Everingham et al., 2010) (middle) and Painting from DomainNet (Peng et al., 2019) (right). Those populate the extended dataset $\mathcal{D}_{sd}$ for distillation.

performance, even for a variant that only mixes images of the same class (see Section 4.3). This supports the intuition that the generated images are diverse enough to potentially extend beyond the scope of each class, while remaining close enough to the overall training domain to still be useful for distillation.

# 4 Experiments

We evaluate our distillation protocol across three families of tasks: classification on various domains, fine-grained classification, and semantic segmentation. For classification, we consider the painting, sketch and clipart datasets from DomainNet (Peng et al., 2019), each composed of the same 345 classes, for which we isolate 20% of the training set for testing. Fine-grained classification is conducted on the CUB (Wah et al., 2011), FGVC Aircraft (Maji et al., 2013) and DTD (Cimpoi et al., 2014) datasets respectively consisting of 200 bird species, 100 aircraft models, and 47 textures. Finally, we use three benchmarks for segmentation: ADE20K (Zhou et al., 2017), Cityscapes (Cordts et al., 2016), and the augmented Pascal VOC (Everingham et al., 2010). After an overview of our experimental setup (Section 4.1), we present our main distillation results (Section 4.2) followed by additional ablation studies (Section 4.3).

## 4.1 Experimental setting

We present the design choices for our student and teacher models and detail the data augmentation applied, the training hyperparameters and our evaluation protocol.

**Backbone models.** For the teacher, we start from one of the pretrained models provided by DINOv2 (Oquab et al., 2024), either ViT-S, ViT-L or ViT-g, three architectures of increasing capacity. Note that the ViT-L and ViT-S models provided by DINOv2 are distilled from their ViT-g. The teacher is then one of these pretrained models probed for the target downstream task (see top part of Figure 2). We also consider a finetuned ViT-L teacher in our study to investigate the impact of finetuning versus probing strategies for the teacher. We do not explore finetuning the ViT-g model, in line with the paper's focus on maximizing the utility of pretrained models within constraints of limited computational resources.

For the student, we explore two lightweight architectures. The majority of our experiments use a ViT-S model initialized with DINOv2's pretrained weights. We also show that our observations generalize to randomly initialized models: we report experiments with a ResNet-50 model for classification and a DeepLabv3 model (Chen et al., 2017) with ResNet-50 backbone for segmentation.

**Prediction head.** We use a MLP head for classification (unlike DINOv2 which evaluates with a linear head) and DINOv2's linear head for segmentation, for students and for teachers. Note that there is no prediction head for the ResNet-50 and DeepLabv3 models as those are trained from scratch.

When available, the input for the prediction head is defined as follows. In classification tasks, we adhere to DINOv2's process: i) we concatenate the CLS tokens from up to the last four blocks (choosing 4 for DomainNet and 3 for fine-grained tasks), ii) optionally, we concatenate the average pooling of the patch embeddings from the last block (which we only do for DomainNet). For segmentation tasks, we adopt DINOv2's linear evaluation protocol, directly evaluating from the patch embeddings of the last block.

**Synthetic image generation with Stable Diffusion.** We generate synthetic datasets with $n$ times more images than the original training set, setting $n$ to 5 for DomainNet and segmentation tasks, and 10 for the relatively smaller fine-grained classification datasets. As noted earlier, this data augmentation strategy may be viewed as a variant of Mixup (Zhang et al., 2018), akin to interpolating between random pairs of images using the ImageMixer method proposed by Pinkney (2022).

**Standard data augmentation.** In all experiments, we apply classical data augmentation to both the original training images and the synthetic images. For classification tasks involving transformers, we use RandomResizedCrop, ColorJitter, and Mixup, while for ResNet-50, we use TrivialAugment (Müller & Hutter, 2021). Note that Mixup is excluded for synthetic images obtained from ImageMixer, which is already a variant of Mixup based on Stable Diffusion. For segmentation tasks, we adopt the same augmentations as DINOv2 (Oquab et al., 2024)—see details in the appendix. Following the recommendation of Beyer et al. (2022), the student and teacher models receive exactly the same batch of images, transformed with the same data augmentation.

**Training hyperparameters.** Probing runs for 20 epochs for ViT-L/g and 30 epochs for ViT-S, while finetuning lasts for 50 epochs for ViT-L and 80 epochs for ViT-S. We use the AdamW optimizer for training ViTs and SGD with momentum for ResNet-50, and a cosine scheduler in both cases. The selection of weight decay and learning rate is determined through a grid search on the validation set, with specific details available in the appendix. In instances where no predefined validation set exists, we allocate 10% of the training set for this purpose. We use a fixed distillation temperature of $T = 2$ and a constant weighting between $\mathcal{L}_{\text{task}}$ and $\mathcal{L}_{\text{distill}}$ set to $\alpha = 0.5$ for all experiments.

**Evaluation.** We report results averaged over three independent runs with different random seeds. For distillation evaluations, we consider three different teachers, each from independent runs, and conduct 2 runs per teacher for DomainNet and 3 runs for fine-grained and segmentation tasks.

**About our probing results.** We remind that for classification, we use a MLP head while DINOv2 (Oquab et al., 2024) uses a linear head. Please also note that for segmentation, we use an image size of $560 \times 560$ pixels while DINOv2 uses $512 \times 512$. This explains why our probing results are slightly higher than those reported by Oquab et al. (2024) (comparison in the appendix).

## 4.2 Experimental results

We explore distillation with two different students: i) DINOv2's ViT-S pretrained with task-agnostic distillation, and ii) randomly initialized models: a ResNet-50 for classification and a DeepLabv3 with ResNet-50 backbone for segmentation.

Our results are presented in Tables 1 to 3. Table 1 reports probing and finetuning results for DINOv2 ViT-S, ViT-L and ViT-g pretrained models. Table 2 reports distillation results using ViT-S as the student, and using a probed ViT-S, a probed and a finetuned ViT-L or a probed VIT-g as the teacher. Table 3 reports distillation results using a probed ViT-g as the teacher, and a randomly initialized ResNet-50 as the student. Distillation results reported in Tables 2 and 3, both with and without augmenting the training set with synthetic images for distillation, are compared to those obtained with a simple probing or finetuning of the

| Model | | | Classification on DomainNet (acc) | | | Fine-grained classification (acc) | | | Semantic segmentation (mIoU) | | |
|---|---|---|---|---|---|---|---|---|---|---|---|
| | | | Painting | Sketch | Clipart | CUB | Aircraft | DTD | ADE20K | Cityscapes | VOC |
| ViT-S | *(1a)* Probing | | 77.3 | 71.9 | 79.3 | 88.2 | 77.1 | 82.1 | 45.1 | 67.0 | 81.8 |
| | *(1b)* Finetuning | | 79.4 | 76.0 | 81.8 | 87.3 | 87.8 | 81.6 | 49.8 | 75.8 | 84.6 |
| ViT-L | *(2a)* Probing | | 82.9 | 80.4 | 85.3 | 91.3 | 87.8 | 85.5 | 47.8 | 70.4 | 82.7 |
| | *(2b)* Finetuning | | 83.9 | 81.4 | 85.9 | 91.5 | 94.0 | 85.8 | 57.4 | 78.6 | 88.0 |
| ViT-g | *(3a)* Probing | | 83.0 | 81.2 | 85.7 | 91.6 | 88.1 | 85.8 | 48.8 | 71.2 | 83.5 |

Table 1: **Probing/finetuning of DINOv2 pretrained models** for classification on DomainNet, fine-grained classification and semantic segmentation. We report accuracy for classification and mIoU for segmentation. Relative distillation gains in Table 2 are with respect to underlined results in this table.

| Student | Teacher | | SD | Classification on DomainNet (acc) | | | Fine-grained classification (acc) | | | Semantic segmentation (mIoU) | | |
|---|---|---|---|---|---|---|---|---|---|---|---|---|
| | | | | Painting | Sketch | Clipart | CUB | Aircraft | DTD | ADE20K | Cityscapes | VOC |
| ViT-S | ViT-S probed | *(4a)* | ✗ | 80.0 (+0.6) | 76.9 (+0.9) | 82.2 (+0.4) | 89.4 (+1.7) | 86.5 (-1.3) | 82.9 (+0.8) | 49.6 (-0.2) | 71.2 (-4.6) | 84.6 (+0.0) |
| | | *(4b)* | ✓ | 80.2 (+0.8) | 77.1 (+0.2) | 82.4 (+0.6) | 89.7 (+1.5) | 86.6 (-1.2) | 83.4 (+1.3) | 50.3 (+0.5) | 72.3 (-3.5) | 84.9 (+0.3) |
| | ViT-L probed | *(5a)* | ✗ | 80.5 (+1.1) | 77.8 (+1.8) | **83.4** (+1.6) | 89.7 (+1.5) | 89.2 (+1.4) | 83.4 (+1.3) | 50.7 (+0.9) | 74.0 (-1.8) | 85.5 (+0.9) |
| | | *(5b)* | ✓ | **80.8** (+1.4) | **78.0** (+2.0) | 83.2 (+1.4) | **90.0** (+1.8) | **89.8** (+2.0) | **84.0** (+1.9) | **51.7** (+1.9) | 74.7 (-1.1) | **86.1** (+1.5) |
| | ViT-L finetuned | *(6a)* | ✗ | 79.7 (+0.3) | 77.0 (+1.0) | 82.5 (+0.7) | 88.6 (+0.4) | 88.9 (+1.3) | 81.5 (-0.6) | 50.7 (+0.9) | **76.3** (+0.5) | 84.8 (+0.2) |
| | | *(6b)* | ✓ | 80.3 (+0.9) | 77.2 (+1.2) | 82.9 (+1.1) | 88.6 (+0.4) | 89.1 (+1.5) | 82.5 (+0.4) | 51.6 (+1.8) | **76.4** (+0.6) | 85.7 (+1.1) |
| | ViT-g probed | *(7a)* | ✗ | 80.5 (+1.1) | 77.7 (+1.7) | **83.4** (+1.6) | 89.1 (+0.9) | **89.6** (+1.8) | 83.1 (+1.0) | 51.6 (+1.8) | 74.4 (-1.4) | 85.7 (+1.1) |
| | | *(7b)* | ✓ | **80.8** (+1.4) | **78.0** (+2.0) | 83.3 (+1.5) | **89.8** (+1.6) | **90.1** (+2.3) | **83.6** (+1.5) | **52.1** (+2.3) | 75.0 (-0.8) | **86.3** (+1.7) |

Table 2: **Distillation on ViT-S initialized with DINOv2** for classification on DomainNet, fine-grained classification and semantic segmentation. We report accuracy for classification and mIoU for segmentation. We report results with and without data augmentation based on Stable Diffusion (SD), for various choices of teachers. Relative gains with respect to simple probing or finetuning (best underlined in Table 1) are in parentheses. **Bold** numbers: within 95% confidence interval of the best score for each task.

| Student | Teacher | | SD | Classification on DomainNet (acc) | | | Fine-grained classification (acc) | | | Semantic segmentation (mIoU) | | |
|---|---|---|---|---|---|---|---|---|---|---|---|---|
| | | | | Painting | Sketch | Clipart | CUB | Aircraft | DTD | ADE20K | Cityscapes | VOC |
| R50 | - | *(8a)* | - | 66.0 | 68.1 | 72.5 | 73.3 | 85.0 | 63.5 | 37.8 | **67.9** | 67.5 |
| | ViT-g probed | *(9a)* | ✗ | 67.7 (+1.3) | 70.5 (+2.4) | **74.9** (+2.4) | 76.0 (+2.7) | 85.7 (+0.7) | 66.7 (+3.2) | 38.2 (+0.4) | **67.7** (-0.2) | 67.7 (+0.2) |
| | | *(9b)* | ✓ | **69.1** (+2.7) | **71.0** (+2.9) | **75.2** (+2.7) | **79.1** (+5.8) | **87.8** (+2.8) | **69.4** (+5.9) | **42.1** (+4.3) | **69.3** (+1.4) | **73.9** (+6.2) |

Table 3: **Distillation from ViT-g to ResNet-50 (resp. DeepLabv3-ResNet50 for segmentation) trained from scratch** for classification on DomainNet, fine-grained classification and semantic segmentation. We report accuracy for classification and mIoU for segmentation. We report results with and without data augmentation based on Stable Diffusion (SD). Relative gains with respect to simple training (underlined) are in parentheses. **Bold** numbers: within 95% confidence interval of the best score for each task.

ViT-S (best underlined in Table 1) and with simple training of ResNet-50 (underlined in Table 3), with relative gains indicated in parentheses.

We discuss the results of Tables 1 to 3 according to four separate axes supporting the main claims of this study: i) the relative gains of distillation over finetuning when teaching a small pretrained model, ii) the impact of finetuning the teacher, iii) the impact of using a teacher that is less accurate than the student, and iv) the generalization of our observations to students trained from scratch.

In what follows, observations are discussed by comparing lines of Tables 1 to 3. The lines from these three tables are denoted by unique alphanumeric reference such that *e.g.* the mention *(2a vs 2b)* refers to comparing lines *2a* and *2b* in the Table 1.

**Task-specific distillation complements task-agnostic distillation.** The key observation from Table 2 is that *task-specific distillation generally outperforms probing and finetuning.* This can be observed by comparing any line from Table 2, *4* to *7*, *a* or *b*, with the corresponding number in line *1* from Table 1, referred to as *(4-7 vs 1)* following the notation introduced in the previous paragraph. Figure 4 illustrates this observation on ADE20K: the PCA of patch embedding representations exhibits a better clustering structure

Teacher        Student init.        Student finetuned        Student distilled

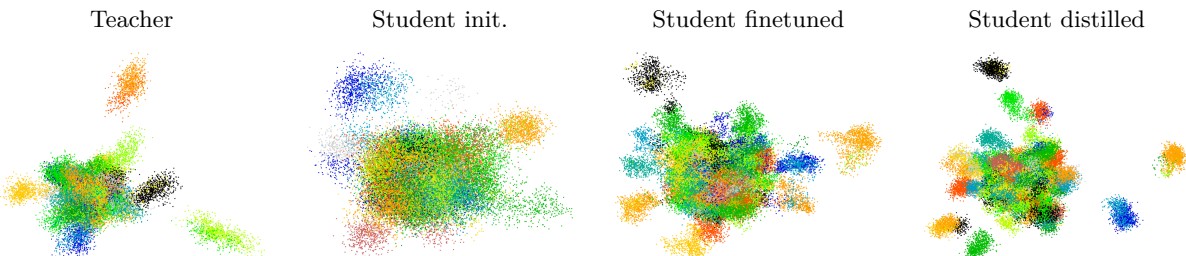

Figure 4: PCA of patch embedding representations for 20 classes of ADE20K for the ViT-g teacher (a) and for the ViT-S student in its initial state (b), after finetuning (c) and after distillation (d), colored by their main class (details in the appendix). Classes are better clustered after distillation than after finetuning.

after distillation than after finetuning (see also Figure 1). Cityscapes is the only exception where distillation from a probed ViT-L or a probed ViT-g does not improve over finetuning. Interestingly, on Cityscapes, the finetuned ViT-S student already outperforms the probed ViT-g and ViT-L teachers by a large margin (+4.6 and +5.4 mIoU) *(1b vs 3a,2a)*, which may explain why distilling from those is not beneficial.

Our dataset augmentation based on Stable Diffusion further enhances distillation results (*4a vs 4b, 5a vs 5b*, etc.), except on Clipart, where it performs on par with distillation on the original training images alone.

While ViT-g exhibits slightly higher accuracy than ViT-L when probed on downstream tasks *(3a vs 2a)*, both models serve as almost equally effective teachers for distillation *(7 vs 5)*. In exploring experiments with smaller teachers, we evaluate distillation from a probed ViT-S *(1a)*, placing ourselves in the context of self-distillation. We observe that when the performance gap between probing and finetuning is not too large, self-distillation *(4)* improves over finetuning *(1b)*, evident across all baselines except for Aircraft and Cityscapes, where the probed ViT-S has an accuracy/mIoU approximately 10% lower than with finetuning *(1a vs 1b)*. However, it is important to note that ViT-g and ViT-L remain superior teachers compared to ViT-S. This implies that, even if ViT-S was pretrained with generic distillation from ViT-g, *it is more effective to directly leverage the largest teachers for downstream tasks.*

**Finetuning yields a poorer teacher than probing.** Next, we study the impact of finetuning our teacher prior to distillation, comparing distillation results using either a probed or finetuned pretrained ViT-L model from DINOv2 (Oquab et al., 2024). Finetuning significantly enhances ViT-L's accuracy compared to probing *(2a vs 2b)*. However, employing the finetuned ViT-L model as a teacher generally results in a poorer performance for the student *(5 vs 6)*. For example, finetuning brings about 6% increase in accuracy for Aircraft and Pascal VOC compared to probing *(2a vs 2b)*, yet the distillation results with the probed teacher are better *(5 vs 6)*. This suggests that *preserving the rich representations learned during pretraining is crucial, even if it leads to a teacher with lower accuracy for the specific task.* Cityscapes is the only exception where distillation of a finetuned teacher significantly improves results, while using a probed teacher degrades them *(5 vs 6)*. This may be attributed to the substantial performance gap between probing and finetuning on this dataset, with +8 to +9 in mIoU *(1a vs 1b, 2a vs 2b)*.

In summary, our experiments indicate that *finetuning the teacher for task-specific distillation is often unnecessary and sometimes even detrimental.* Given the relatively fast training of the MLP head, the primary computational cost in knowledge distillation with teacher probing lies in training the student, with the additional overhead of performing forward passes through the teacher.

**Teachers are not required to be as accurate as students.** Sometimes, simply finetuning our ViT-S model *(1b)* gives better results than probing ViT-g *(3a)*. This is the case for all three segmentation tasks. Still, distilling from ViT-g proves beneficial for ADE20K and Pascal VOC *(7b)*, as it gives around 2% mIoU gain compared to finetuning *(1b)*, even though the finetuned ViT-S model is already about 1% higher in mIoU than the teacher *(3a)*. This supports the more general observation that *a student can surpass its teacher and still benefit from distillation.*

| | Text prompt-free | Student: ViT-S | | | Student: ResNet-50 | | |
|---|:---:|:---:|:---:|:---:|:---:|:---:|:---:|
| | | CUB | Aircraft | DTD | CUB | Aircraft | DTD |
| Baseline (distillation only from $\mathcal{D}_{\text{train}}$) | ✓ | 89.1 | 89.6 | 83.1 | 76.0 | 85.7 | 66.7 |
| $\mathcal{D}_{\text{sd}}$ uses Text-to-image (parent class) | ✗ | 89.4 | 90.0 | 83.6 | 77.8 | 87.9 | 68.1 |
| $\mathcal{D}_{\text{sd}}$ uses Text-to-image (class name) | ✗ | 89.5 | 90.2 | 83.5 | 79.8 | 87.6 | 68.7 |
| $\mathcal{D}_{\text{sd}}$ composed of Random ImageNet images | ✓ | 89.2 | 89.4 | 83.3 | 77.3 | 86.5 | 65.8 |
| $\mathcal{D}_{\text{sd}}$ uses ImageVariations | ✓ | 89.5 | 90.4 | 83.4 | 78.8 | 87.7 | 68.7 |
| $\mathcal{D}_{\text{sd}}$ uses **ImageMixer** | ✓ | 89.8 | 90.1 | 83.6 | 79.1 | 87.8 | 69.4 |

Table 4: **Building $\mathcal{D}_{\text{sd}}$ from $\mathcal{D}_{\text{train}}$.** We compare distillation results, using ViT-g as a teacher and ViT-S or ResNet-50 as a student, for our mixing approach based on stable diffusion (ImageMixer, by Pinkney 2022) with i) a model producing image variations from single images (ImageVariations, by Pinkney 2022), ii) simply adding a subset of ImageNet, and iii) text-to-image diffusion using the parent class only, *i.e.* bird, aircraft or texture ("A photo of a {parent class}"), or using class information as well ("A photo of a {class name} {parent class}"). We observe that the ImageMixer variant we advocate for is surprisingly competitive despite not requiring a text prompt.

**Data augmentation based on Stable Diffusion substantially helps students trained from scratch.** Here, we replace DINOv2's pretrained ViT-S student with a ResNet-50 (resp. DeepLabv3 with a ResNet-50 backbone for segmentation) trained from scratch, while retaining DINOv2's pretrained ViT-g as the teacher. Results are reported in Table 3 and consistently demonstrate that distillation is beneficial, leading to 2% accuracy gain on average. Notably, data augmentation based on Stable Diffusion significantly enhances results, yielding a further 2-3% accuracy gain on fine-grained tasks and a 4-6% mIoU gain for segmentation compared to standard distillation *(9a vs 9b)*. Surprisingly, the ResNet-50 model benefits even more from distillation than the pretrained ViT-S model. These findings indicate that the observations made for a pretrained ViT-S student generalize to students that i) did not undergo generic distillation or any form of pretraining, and ii) whose architecture is not based on transformers like the teacher.

### 4.3 Ablation studies

**Data augmentation with Stable Diffusion.** We now compare various strategies for creating an augmented dataset $\mathcal{D}_{\text{sd}}$ used for distillation. Our evaluation focuses on fine-grained classification tasks, involving both a pretrained ViT-S and a ResNet-50 trained from scratch as students.

We conduct a comparative analysis of our data augmentation strategy based on ImageMixer (Pinkney, 2022). We compare it to: i) another model by Pinkney (2022) creating image variations from single images; ii) an augmentation approach incorporating an ImageNet subset; and iii) the text-to-image diffusion model used by Sarıyıldız et al. (2023). For the latter, we explore textual prompts with the parent class, and with and without class names. Specifically, prompts with class names take the form "A photo of a {class name} {parent class}" while prompts without class names follow the pattern "A photo of a {parent class}", where {parent class} represents either bird, aircraft, or texture.

Table 4 shows that our prompt-free approach performs equally well, if not better, than a prompt-based data augmentation that leverages class information. Additionally, prompt engineering poses challenges for certain tasks, especially segmentation, and necessitates the model used for parsing prompts (*e.g.*, CLIP) to be trained with semantic information about the data. This may be impossible for some modalities such as medical or microscopy images, which are not easily described with text and might fall outside the semantic scope expected by CLIP-like models.

Our chosen strategy based on synthetic images produced using the ImageMixer model outperforms the ImageVariations model on 5 out of 6 settings, validating the benefits of a mixing-based approach conditioning image generation with multiple images.

| | Data used in .. | | CUB | Aircraft | DTD |
|---|---|---|---|---|---|
| | $\mathcal{L}_{\text{distill}}$ | $\mathcal{L}_{\text{task}}$ | | | |
| Finetuning | - | $\mathcal{D}_{\text{sd-intra}}$ | 83.8 | 85.4 | 80.3 |
| | | $\mathcal{D}_{\text{train}}$ | 87.3 | 87.8 | 81.6 |
| Distillation | $\mathcal{D}_{\text{sd-intra}}$ | $\mathcal{D}_{\text{sd-intra}}$ | 89.6 | 89.0 | 83.6 |
| | $\mathcal{D}_{\text{sd-intra}}$ | $\mathcal{D}_{\text{train}}$ | 89.6 | 90.1 | 83.9 |
| | $\mathcal{D}_{\text{sd}}$ | $\mathcal{D}_{\text{train}}$ | 89.8 | 90.1 | 83.6 |

Table 5: **Impact of synthetic data on each loss.** Impact on fine-grained classification tasks for finetuning and distillation with ViT-S as student and ViT-g as teacher, using a dataset $\mathcal{D}_{\text{sd-intra}}$ augmented with synthetic images by mixing original images inside each class separately.

| | Loss | SD | CUB | Aircraft | DTD |
|---|---|---|---|---|---|
| Finetuning | $\mathcal{L}_{\text{train}}$ | - | 87.3 | 87.8 | 81.6 |
| Distillation | $\mathcal{L}_{\text{distill}}$ | ✗ | 88.8 | 86.2 | 82.7 |
| | | ✓ | 89.6 | 86.9 | 82.9 |
| | $\mathcal{L}_{\text{train}}+\mathcal{L}_{\text{distill}}$ | ✗ | 89.1 | 89.6 | 83.1 |
| | | ✓ | 89.8 | 90.1 | 83.6 |

Table 6: **Role of the different losses**. We compare optimizing $\mathcal{L}_{\text{train}}$ only (*i.e.* finetuning), $\mathcal{L}_{\text{distill}}$ only, or both losses with equal weighting (standard distillation followed in our experiments). Results are with ViT-S as student and ViT-g as teacher.

**Using augmented data for supervision.** In our study, we use synthetic data only for optimizing the distillation loss $\mathcal{L}_{\text{distill}}$, while the task-specific loss $\mathcal{L}_{\text{task}}$ is trained solely on real data. In this section, we explore the outcomes when incorporating synthetic images as additional labeled data for optimizing $\mathcal{L}_{\text{train}}$, for both finetuning and distillation. For this purpose, we compare two different ways of leveraging the diffusion model of Pinkney (2022) described in Section 3: mixing images regardless of their labels (inter-class), or mixing images from each class separately (intra-class). These approaches result in two augmented datasets, $\mathcal{D}_{\text{sd}}$ and $\mathcal{D}_{\text{sd-intra}}$, containing both the original images and the synthetic ones, as explained in Section 3. Table 5 presents distillation results for the fine-grained datasets. We observe comparable performance between inter-class and intra-class approaches, but incorporating synthetic data for supervision is not beneficial. In particular, including synthetic images for finetuning considerably degrades results. This aligns with the intuition that the diffusion model may not be faithful enough to each fine-grained class, and even when provided with two images of the same class, it may generate a new image beyond the scope of this class.

**Relative weighting between task and distillation losses.** Here, we investigate the influence of completely excluding the loss $\mathcal{L}_{\text{task}}$ during distillation. Table 6 presents the results for fine-grained classification when solely optimizing $\mathcal{L}_{\text{train}}$ (*i.e.*, finetuning), solely optimizing $\mathcal{L}_{\text{distill}}$, and optimizing both with equal weighs, as implemented in our study. The outcomes reveal that training without label information (*i.e.*, optimizing $\mathcal{L}_{\text{distill}}$ only) yields competitive results for CUB but significantly lower results for Aircraft. Overall, the student achieves the best results when exposed to both hard labels and soft teacher labels.

## 5  Discussion and concluding remarks

Since the seminal work of Sharif Razavian et al. (2014), it has been known that generic pretrained models could be reused directly or adapted for many target tasks instead of learning new models from scratch. Yet, the rapid development and public release of even larger, rich and generic models, pretrained on up to billions of images (Oquab et al., 2024; Fang et al., 2023), raises a pressing question with heavy practical implications: *How to best leverage the knowledge of large and generic visual models when training a smaller model for a specific task?* Our work aims at addressing this question by reexamining current good practices for knowledge distillation in the light of these new large models, and draws a series of experimental conclusions. Below we summarize the main messages of our study, and relate them to previous discussions on neighboring topics.

**Task-specific distillation and self-supervised learning.** In the context of self-supervised learning, knowledge distillation has emerged as a compelling way to compress large pretrained models into smaller ones, yielding significant improvements compared to directly pretraining these small models (Abbasi Koohpayegani et al., 2020; Fang et al., 2021; Xu et al., 2022; Wu et al., 2022; Oquab et al., 2024). Yet, our study shows that simply finetuning or probing these small pretrained models yields sub-optimal results compared to leveraging the knowledge of the larger models for a specific downstream task.

**Accurate teachers may not be the best for distillation.** In prior works, Cho & Hariharan (2019); Mirzadeh et al. (2020) have observed that the most accurate teachers are not always the best for distillation, attributing this observation to the model capacity gap between the student and the teacher. To make the most of the largest models, Mirzadeh et al. (2020) propose a multi-stage approach where knowledge is distilled from a large model to successively smaller ones, thus reducing the capacity gap between two successive distillation steps. Pointing to the inefficiency of such approach, Cho & Hariharan (2019) show that instead, this capacity gap can be mitigated by stopping the teacher's training early. This early-stopping approach may be related to our observation: by freezing the teacher's pretrained backbone, *i.e.*, *probing* the model instead of *finetuning* it, we prevent it from specializing too much to the given task. This results in better distillation despite a lower teacher accuracy. Nevertheless, we did not find any evidence that a large capacity gap may be detrimental to distillation, since our best results were obtained by distilling directly from DINOv2's (Oquab et al., 2024) largest ViT-g model to much smaller models, such as ViT-S or ResNet-50. Instead, the fact that a probed model can serve as a better teacher than a finetuned one suggests that some aspects of the representation that are still relevant to the task are lost when training for that task. This phenomenon is further discussed in the next paragraph.

**Probing can yield better teachers than finetuning.** One of the key observations of our study is that, at comparable performance (experimentally, a difference in accuracies smaller than 6% for the three families of models considered here, DINOv2, EVA-02 and EVA-02-CLIP), a probed model makes a better teacher than a finetuned one. This observation could be due to the *catastrophic forgetting* which happens (Kirkpatrick et al., 2017) when performing finetuning. Our experiments suggest that, when specializing for a given task, the finetuned teacher has forgotten some features that are not immediately relevant for optimizing the task loss, but that still help generalization. For instance, when finetuned on the CUB bird classification task, the pretrained model could end up only relying on *spurious correlations* (such as the background, a standard source of spurious correlations in CUB, as studied in Sagawa et al. 2019) that provide *shortcuts* to the optimization. These shortcuts help improving the teacher accuracy but are detrimental to the distillation process.

**A teacher need not be more accurate than its student.** Prior works have showed that student models can learn from poorly trained teachers (Yuan et al., 2020), or teachers with the same architecture (Furlanello et al., 2018), sometimes even outperforming them. Our results are consistent with these findings, since we also observed that a small model can benefit from distillation even when that same model already outperforms its larger teacher after simple finetuning. Additionally, distillation also resulted in improvements when using a teacher of the same size as the student (*i.e.*, self-distillation). However, our results highlight that distilling from the largest models works considerably better than self-distillation, thus supporting the idea that knowledge from larger models can further guide the training of a smaller student.

**Stable Diffusion as a source of additional information.** Our study shows that our data augmentation strategy producing synthetic images with Stable Diffusion can be leveraged to extend the set of images used to optimize the distillation loss $\mathcal{L}_{\text{distill}}$. However, as illustrated in Table 5, including these synthetic images in the optimization of the task loss $\mathcal{L}_{\text{task}}$ can degrade results, particularly for finetuning. This shows that it may be difficult to efficiently leverage this augmentation strategy outside the context of knowledge distillation. Using synthetic image-label pairs for supervised learning requires i) the generation process to be good enough for images to pertain to their class, which may be challenging for fine-grained tasks, and ii) the generation process to create the right label for each new image, which is particularly challenging for dense tasks such as semantic segmentation. Knowledge distillation alleviates the need for generating class-specific images and for labeling generated images, and hence can more easily leverage Stable Diffusion.

**Acknowledgments**

This project was supported by ANR 3IA MIAI@Grenoble Alpes (ANR-19-P3IA-0003) and by ERC grant number 101087696 (APHELEIA project). This work was granted access to the HPC resources of IDRIS under the allocation [AD011013343R1] made by GENCI.

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

# Appendix

In this appendix, we first introduce additional experimental details regarding the choice of training hyperparameters and data augmentation, the prediction heads, PCA visualizations and training time (Appendix A). Next, we provide additional experimental results, with a comparison of linear and MLP heads for classification, an extended version of Tables 1 to 3 with confidence intervals, additional distillation results with EVA-02 MIM- and CLIP-pretrained models (Fang et al., 2023; Sun et al., 2023) instead of DINOv2, and a comparison of our chosen distillation loss (Hinton et al., 2015) to alternatives from the literature (Appendix B). Finally, we include additional visualizations of synthetic images produced by our mixing based on Stable Diffusion (Appendix C).

## A    Additional experimental details

### A.1    Datasets

Table A reports the number of classes and the number of images in the training set of the datasets used for our study.

| | | Classes | Size (train) |
|---|---|---|---|
| DomainNet(Peng et al., 2019) | Painting | 345 | 60617 |
| | Sketch | | 56304 |
| | Clipart | | 39064 |
| Fine-grained classification | CUB (Wah et al., 2011) | 200 | 5994 |
| | Aircraft (Maji et al., 2013) | 100 | 6667 |
| | DTD (Cimpoi et al., 2014) | 47 | 3760 |
| Semantic segmentation | ADE20K (Zhou et al., 2017) | 150 | 20210 |
| | Cityscapes (Cordts et al., 2016) | 19 | 2975 |
| | Pascal VOC (aug.) (Everingham et al., 2010) | 21 | 10582 |

Table A: Number of classes and size of training set of each dataset.

### A.2    Training hyperparameters

Table B details the weight decay and learning rate used for each task (classification on DomainNet, fine-grained classification, semantic segmentation), each architecture, and each training procedure (with/without freezing the pretrained backbone). Values are chosen based on a grid search on the validation set. More precisely, a coarse grid search is first performed on a logarithmic scale using powers of 10, before defining a finer one that is reported in Table B. When the grid search leads to values that are nearly identical for all tasks, we fix the value and report it in the table. Note that distillation with synthetic images based on Stable Diffusion is run with the best hyperparameters found for distillation without synthetic images. Also note that for finetuning experiments on ViT-L, we use a smaller batch size (8 for segmentation, 32 for classification) and reduce the learning rate in the grid search accordingly.

### A.3    Generic data augmentation

In all experiments, we consistently apply classical data augmentation to both training and synthetic images (except for Mixup which is only applied to original images). The list of augmentations with their parameters is detailed below for each task (classification or segmentation) and architecture.

**Classification.**    When training ViTs, we apply:

| | Arch | | Learning rate | Weight decay |
|---|---|---|---|---|
| DomainNet | ViT | Probing
Finetuning/distillation | $\{.0001, .0002, .0004\}$
$\{1, 2, 4\} \times 10^{-5}$ | $0$
$\{.025, .05, .1\}$ |
| | ResNet-50 | Training/distillation | $.1$ | $.0005$ |
| Fine-grained
classification | ViT | Probing
Finetuning/distillation | $\{.001, .002, .004\}$
$\{1, 2, 4\} \times 10^{-5}$ | $\{.5, 1, 2, 4, 8\}$
$\{.025, .05, .1\}$ |
| | ResNet-50 | Training/distillation | $\{.01, .02, .04\}$ | $\{.005, .01\}$ |
| Semantic
segmentation | ViT | Probing
Finetuning/distillation | $.008$
$\{1, 2, 4\} \times 10^{-5}$ | $0$
$\{.001, .01, .1\}$ |
| | DeepLabv3(R50) | Training/distillation | $\{.01, .02, .04, .08\}$ | $\{.0001, .001, .01\}$ |

Table B: Training hyperparameters for a batch size of 128 for DominNet and fine-grained classification tasks and 16 for segmentation tasks (32 for probing). Hyperparameters in {} are chosen based on a grid search on the validation set.

- RandomResizedCrop with scale 0.08

- ColorJitter with range $(0, 0.4)$

- RandomFlip with probability 0.5

- Mixup with parameter 0.2.

An exception is for probing on fine-grained classification tasks, where we simply apply Resize and CenterCrop instead of RandomResizedCrop, and do not apply Mixup. We found that these transformations were too strong for fine-grained classification with a frozen backbone.

When training the ResNet-50, we use TrivialAugment's (Müller & Hutter, 2021) strategy (ImageNet version), that consists of

- RandomResizedCrop with scale 0.08

- ColorJitter with range $(0, 0.4)$

- RandomFlip with probability 0.5

- A fourth transformation randomly sampled among a pool.

However for fine-grained classification, we use a scale parameter of 0.4 for RandomResizedCrop, as 0.08 proved too strong for training from scratch on fine-grained tasks.

For validation and testing, we follow the standard procedure of applying Resize and CenterCrop.

**Semantic segmentation.** We train with images of size $(s, s)$ with $s = 560$. We apply the following data augmentations for all experiments, which correspond to the *mmsegmentation* augmentations also used by DINOv2:

- Resize to $(., s)$ with ratio range $(0.5, 2.0)$

- RandomCrop to $(s, s)$, with cat_max_ratio $= 0.75$

- RandomFlip with probability 0.5

- PhotoMetricDistortion.

For validation and testing, we use sliding windows of size $(s, s)$ and stride $\frac{s}{2}$.

### A.4 Details on prediction heads

We remind the reader that for segmentation, we use the same linear evaluation head as DINOv2's (Oquab et al., 2024) while for classification, we use a MLP head, unlike DINOv2 which uses a linear head.

More precisely, let $n_{\text{in}}, n_{\text{hidden}}, n_{\text{out}}$ respectively denote the number of input, hidden, and output neurons in the MLP head. $n_{\text{out}}$ is the number of classes, and $n_{\text{in}}$ the number of input features extracted from the pretrained backbone, meaning that $n_{\text{in}} = n_f \times (n_{\text{CLS}} + \mathbf{1}_{\text{use avgpool}})$, where

- $n_f$ is the embedding dimension ($n_f = 1536, 1024, 384$ for ViT-g, ViT-L and ViT-S respectively)

- $n_{\text{CLS}}$ is the number of blocks from which the CLS tokens are concatenated ($n_{\text{CLS}} = 4$ for DomainNet, 3 for fine-grained tasks)

- $\mathbf{1}_{\text{use avgpool}}$ indicates whether we also concatenate the average pooling of the patch embeddings of the last block (true for DomainNet).

As for the number of hidden neurons $n_{\text{hidden}}$, we set $n_{\text{hidden}} = n_{\text{in}}$ for ViT-S and $n_{\text{hidden}} = \sqrt{n_{\text{in}} \times n_{\text{out}}}$ for ViT-L/ViT-g, as we experimentally found that this choice gave the best results. Intuitively, using such intermediate size for ViT-L/ViT-g, whose embedding sizes are larger (1024 and 1536), allow for a more progressive decrease toward $n_{\text{out}}$.

### A.5 Details on the PCA

The main paper provides PCA-based visualizations of the learned representations for CUB and ADE20K datasets, respectively in Figure 1 and 4. Detailed step-by-step descriptions of how these visualizations were constructed are provided below.

For the PCA visualization of teacher predictions from Fig. 1 on the CUB fine-grained classification task, the steps are the following:

1. **Feature computation** for both original and synthetic CUB training images, giving class token predictions of shapes $(N, D)$ and $(N_{\text{synthetic}}, D)$ with $D$ the embedding dimension ($D = 1536$ for ViT-g and 384 for ViT-S)

2. **Subsampling**: we only keep the first 20 classes. We keep synthetic images that result from a mix of images belonging to this set of 20 classes. This leaves $M < N$ and $M_{\text{synthetic}} < N_{\text{synthetic}}$ images.

3. **PCA** over the $(M, D)$ predictions on original images

4. **Visualization** of the $(M + M_{\text{synthetic}}, D)$ data points projected onto the two main principal components, colored by class label for the $M$ images, and in gray for the $M_{\text{synthetic}}$ images.

For the visualization of student predictions on Figure 1, the steps are the same but without the synthetic images.

For the PCA visualization from Figure 4 on the ADE20K segmentation task, we visualize patch embedding representations as follows:

1. **Feature computation** on $N = 500$ test images, giving patch embedding predictions of shape $(N, D, H, W)$ with $D$ the embedding dimension; and $H = W = 40$ ($= \frac{\text{image size}}{\text{patch size}} = \frac{560}{14}$).

2. **Resizing** of the corresponding 500 segmentation maps to shape $(N, C, H, W)$ where $C$ is the number of classes. We use *mmsegmentation*'s *resize* method on one-hot encoded labels.

3. **Flattening** of predictions and labels to $(N \times H \times W, D)$, $(N \times H \times W, C)$ respectively.

4. **Filtering**: we keep patches whose labels are well defined, with a probability over 0.9.

| Model | Head | | Classification on DomainNet | | | Fine-grained classification | | | Semantic segmentation | | |
|---|---|---|---|---|---|---|---|---|---|---|---|
| | | | Painting | Sketch | Clipart | CUB | Aircraft | DTD | ADE20K | Cityscapes | VOC |
| ViT-g | Probing | Linear - Oquab et al. (2024) | - | - | - | 91.6 | 87.2 | 84.5 | 49.0 | 71.3 | 83.0 |
| | | Linear - ours | 82.3 | 80.5 | 84.9 | 91.7 | 87.8 | 85.5 | 48.8 | 71.2 | 83.5 |
| | | MLP - outs | 83.0 | 81.2 | 85.7 | 91.6 | 88.1 | 85.8 | - | - | - |
| ViT-L | Probing | Linear - Oquab et al. (2024) | - | - | - | 90.5 | 81.5 | 84.0 | 47.7 | 70.3 | 82.1 |
| | | Linear - ours | 82.2 | 79.9 | 85.1 | 91.8 | 86.5 | 85.4 | 47.8 | 70.4 | 82.7 |
| | | MLP - ours | 82.9 | 80.4 | 85.3 | 91.3 | 87.8 | 85.5 | - | - | - |
| ViT-S | Probing | Linear - Oquab et al. (2024) | - | - | - | 88.1 | 74.0 | 80.6 | 44.3 | 66.6 | 81.1 |
| | | Linear - ours | 75.6 | 70.0 | 78.7 | 89.0 | 75.8 | 80.8 | 45.1 | 67.0 | 81.8 |
| | | MLP - ours | 77.3 | 71.9 | 79.3 | 88.2 | 77.1 | 82.1 | - | - | - |
| | Finetuning | Linear - ours | 78.5 | 75.6 | 81.3 | 87.0 | 87.3 | 81.2 | 49.8 | 75.8 | 84.6 |
| | | MLP - ours | 79.4 | 76.0 | 81.8 | 87.3 | 87.8 | 81.6 | - | - | - |

Table C: **Comparison of our linear probing results with DINOv2's (Oquab et al., 2024) and comparison of linear and MLP heads for classification.** We report accuracy for classification and mIoU for segmentation. We report results from probing ViT-g, ViT-L, ViT-S, and from finetuning ViT-S. The underlined figures correspond to those in Table 1.

5. **Subsampling**: we only keep 20 classes. We select those whose size (number of patches of this class) is closest to the median size.

6. **Filtering** and **subsampling** yield a number $M$ of data points, $M < N \times H \times W$.

7. **PCA** on the $(M, D)$ predictions.

8. **Visualization** of the two main principal components, colored by class label.

## A.6 Training time

All our experiments were performed on a single GPU (either V100 or A100). We detail the training time with a pretrained ViT-g as teacher and a pretrained ViT-S as student. When using a ResNet-50 from scratch as student, the training time per epoch is similar to that of the ViT-S, but we train for longer (200 epochs instead of 80). Experimentally, we observed that probing the teacher (ViT-g) either takes less time or about the same amount of time as finetuning the student (ViT-S). Distillation with the probed ViT-g as teacher takes approximately twice as long as finetuning. Lastly, adding data augmentation based on Stable Diffusion further increases the training time by 1.5 times in average. For example, finetuning the ViT-S for ADE20K takes 16 hours on a A100 GPU, while distillation with data augmentation based on Stable Diffusion takes 55 hours, and probing the ViT-g takes 14 hours.

As for the generation of synthetic data, our image mixing procedure (Pinkney, 2022) roughly takes 2 hours for 1000 images (on a V100 GPU). We remind that we generated synthetic datasets with $n$ times more images than in the training set, with $n = 5$ for DomainNet and semantic segmentation, and $n = 10$ for the relatively smaller fine-grained tasks. The size of the training set of each task is reported in Table A.

## B Additional experimental results

### B.1 Additional results with a linear prediction head

In Table C, we compare classification results using linear and MLP heads when probing ViT-S, ViT-L and ViT-g, and when finetuning ViT-S. We also compare our linear evaluation results with DINOv2's (Oquab et al., 2024). For linear probing, our performance is similar to DINOv2's. Relative differences can be attributed to varying choices of data augmentation for classification, and to differences in image size for segmentation (we train with size 560 while DINOv2 uses 512). For classification, we observe that using a MLP head consistently improves results, both for probing and finetuning. In other words, using a MLP head for both the teacher and student models independently boosts their accuracy, and we experimentally observed that it also makes a better teacher, as it yields the best distillation results for the student.

| Model | | Classification on DomainNet (acc) | | | Fine-grained classification (acc) | | | Semantic segmentation (mIoU) | | |
|---|---|---|---|---|---|---|---|---|---|---|
| | | Painting | Sketch | Clipart | CUB | Aircraft | DTD | ADE20K | Cityscapes | VOC |
| ViT-S | *(1a)* Probing | 77.3 ±.2 | 71.9 ±.3 | 79.3 ±.2 | 88.2 ±.1 | 77.1 ±.4 | 82.1 ±.5 | 45.1 ±.1 | 67.0 ±.2 | 81.8 ±.2 |
| | *(1b)* Finetuning | 79.4 ±.3 | 76.0 ±.2 | 81.8 ±.2 | 87.3 ±.8 | 87.8 ±.9 | 81.6 ±1.1 | 49.8 ±.4 | 75.8 ±.3 | 84.6 ±.7 |
| ViT-L | *(2a)* Probing | 82.9 ±.2 | 80.4 ±.2 | 85.3 ±.1 | 91.3 ±.5 | 87.8 ±1.3 | 85.5 ±.3 | 47.8 ±.2 | 70.4 ±.1 | 82.7 ±.3 |
| | *(2b)* Finetuning | 83.9 ±.2 | 81.4 ±.1 | 85.9 ±.1 | 91.5 ±.1 | 94.0 ±.5 | 85.8 ±.6 | 57.4 ±.1 | 78.6 ±.2 | 88.0 ±.4 |
| ViT-g | *(3a)* Probing | 83.0 ±.4 | 81.2 ±.2 | 85.7 ±.1 | 91.6 ±.2 | 88.1 ±.6 | 85.8 ±.3 | 48.8 ±.2 | 71.2 ±.2 | 83.5 ±.1 |

Table D: **Probing/finetuning of DINOv2 pretrained models** for classification on DomainNet, fine-grained classification and semantic segmentation. We report accuracy for classification and mIoU for segmentation. We report the 95% confidence estimation ($1.96\sigma$) averaged to the upper decimal.

| Student | Teacher | | SD | Classification on DomainNet (acc) | | | Fine-grained classification (acc) | | | Semantic segmentation (mIoU) | | |
|---|---|---|---|---|---|---|---|---|---|---|---|---|
| | | | | Painting | Sketch | Clipart | CUB | Aircraft | DTD | ADE20K | Cityscapes | VOC |
| ViT-S | ViT-S probed | *(4a)* | ✗ | 80.0 ±.2 | 76.9 ±.4 | 82.2 ±.3 | 89.4 ±.3 | 86.5 ±.7 | 82.9 ±.4 | 49.6 ±.2 | 71.2 ±.3 | 84.6 ±.2 |
| | | *(4b)* | ✓ | 80.2 ±.3 | 77.1 ±.2 | 82.4 ±.5 | 89.7 ±.3 | 86.6 ±.6 | 83.4 ±.5 | 50.3 ±.2 | 72.3 ±.3 | 84.9 ±.2 |
| | ViT-L probed | *(5a)* | ✗ | 80.5 ±.3 | 77.8 ±.3 | **83.4** ±.2 | 89.7 ±.4 | 89.2 ±.7 | 83.4 ±.4 | 50.7 ±.3 | 74.0 ±.2 | 85.5 ±.3 |
| | | *(5b)* | ✓ | **80.8** ±.3 | **78.0** ±.2 | 83.2 ±.4 | **90.0** ±.3 | **89.8** ±.4 | **84.0** ±.4 | **51.7** ±.2 | 74.7 ±.2 | **86.1** ±.3 |
| | ViT-L finetuned | *(6a)* | ✗ | 79.7 ±.3 | 77.0 ±.2 | 82.5 ±.3 | 88.6 ±.4 | 88.9 ±.6 | 81.5 ±.7 | 50.7 ±.5 | **76.3** ±.3 | 84.8 ±.4 |
| | | *(6b)* | ✓ | 80.3 ±.2 | 77.2 ±.2 | 82.9 ±.3 | 88.6 ±.3 | 89.1 ±.4 | 82.5 ±.5 | 51.6 ±.7 | **76.4** ±.3 | 85.7 ±.4 |
| | ViT-g probed | *(7a)* | ✗ | 80.5 ±.1 | 77.7 ±.3 | **83.4** ±.2 | 89.1 ±.5 | **89.6** ±.5 | 83.1 ±.8 | 51.6 ±.4 | 74.4 ±.2 | 85.7 ±.3 |
| | | *(7b)* | ✓ | **80.8** ±.2 | **78.0** ±.2 | **83.3** ±.3 | 89.8 ±.4 | **90.1** ±.7 | **83.6** ±.6 | **52.1** ±.4 | 75.0 ±.1 | **86.3** ±.2 |

Table E: **Distillation on ViT-S initialized with DINOv2** for classification on DomainNet, fine-grained classification and semantic segmentation. We report accuracy for classification and mIoU for segmentation. We report results with and without data augmentation based on Stable Diffusion (SD), for various choices of teachers. We report the 95% confidence estimation ($1.96\sigma$) averaged to the upper decimal. **Bold** numbers: within 95% confidence interval of the best score for each task.

| Student | Teacher | | SD | Classification on DomainNet (acc) | | | Fine-grained classification (acc) | | | Semantic segmentation (mIoU) | | |
|---|---|---|---|---|---|---|---|---|---|---|---|---|
| | | | | Painting | Sketch | Clipart | CUB | Aircraft | DTD | ADE20K | Cityscapes | VOC |
| R50 | - | *(8a)* | - | 66.0 ±.4 | 68.1 ±.3 | 72.5 ±.9 | 73.3 ±.2 | 85.0 ±.9 | 63.5 ±1.2 | 37.8 ±.7 | **67.9** ±.7 | 67.5 ±1.0 |
| | ViT-g probed | *(9a)* | ✗ | 67.7 ±.7 | 70.5 ±1.0 | **74.9** ±.4 | 76.0 ±.6 | 85.7 ±.4 | 66.7 ±.6 | 38.2 ±.6 | **67.7** ±1.6 | 67.7 ±.5 |
| | | *(9b)* | ✓ | **69.1** ±.8 | **71.0** ±.3 | **75.2** ±.6 | **79.1** ±.9 | **87.8** ±.5 | **69.4** ±1.1 | **42.1** ±.4 | **69.3** ±1.9 | **73.9** ±.7 |

Table F: **Distillation from ViT-g to ResNet-50 (resp. DeepLabv3-ResNet50 for segmentation) trained from scratch** for classification on DomainNet, fine-grained classification and semantic segmentation. We report accuracy for classification and mIoU for segmentation. We report results with and without data augmentation based on Stable Diffusion (SD). We report the 95% confidence estimation ($1.96\sigma$) averaged to the upper decimal. **Bold** numbers: within 95% confidence interval of the best score for each task.

## B.2   Main results with standard deviations

In Tables D to F, we provide uncertainty estimations to the results reported in Tables 1 to 3 of the main paper, respectively. We use $1.96\sigma$ as 95% confidence estimator, and report its value averaged to the upper decimal, where $\sigma$ is the standard deviation of the results obtained through different runs. We remind that we use 3 runs for probing, finetuning and training, $3 \times 2 = 6$ runs for distillation on DomainNet and $3 \times 3 = 9$ runs for distillation on fine-grained and semantic segmentation tasks.

## B.3   Distillation with EVA-02 pretrained models

In this section, we assess whether the good practices we have drawn as experimental conclusions of our study using DINOv2's pretrained models as teachers also transfer to EVA-02 (Fang et al., 2023) and EVA-02-CLIP (Sun et al., 2023) models, that were pretrained with masked image modeling (MIM) and CLIP training, respectively. More precisely, we use i) either EVA-02 or EVA-02-CLIP ViT-L models as teachers, pretrained

| Model | | Classification on DomainNet | | | Fine-grained classification | | |
|---|---|---|---|---|---|---|---|
| | | Painting | Sketch | Clipart | CUB | Aircraft | DTD |
| EVA-02 ViT-S | *(1a)* Probing | 51.9 | 31.7 | 56.7 | 45.4 | 31.4 | 53.7 |
| | *(1b)* Finetuning | 80.1 | 76.0 | 82.3 | 88.3 | 84.8 | 81.1 |
| EVA-02 ViT-L | *(2a)* Probing | 84.1 | 82.4 | 87.0 | 85.1 | 63.5 | 83.9 |
| | *(2b)* Finetuning | 85.2 | 83.2 | 86.7 | 91.8 | 90.7 | 86.8 |
| EVA-02-CLIP ViT-L | *(2a)* Probing | 83.9 | 82.4 | 86.9 | 85.2 | 63.9 | 83.4 |
| | *(2b)* Finetuning | 85.3 | 83.4 | 86.6 | 91.5 | 93.5 | 86.9 |

Table G: **Probind/finetuning of EVA-02 (Fang et al., 2023) and EVA-02-CLIP (Sun et al., 2023) pretrained models** for classification on DomainNet and fine-grained classification. We report the probing/finetuning accuracies of EVA-02 ViT-L, EVA-02-CLIP ViT-L used as teachers, and EVA-02 ViT-S used as student. Relative distillation gains in Table H are with respect to underlined results in this table.

| Student | Teacher | | SD | Classification on DomainNet | | | Fine-grained classification | | |
|---|---|---|---|---|---|---|---|---|---|
| | | | | Painting | Sketch | Clipart | CUB | Aircraft | DTD |
| EVA-02 ViT-S | EVA-02 ViT-L probed | *(5a)* | ✗ | 81.0 (+0.9) | **78.0** (+2.0) | **83.7** (+1.4) | 87.3 (-1.0) | 78.9 (-5.9) | **83.6** (+2.5) |
| | | *(5b)* | ✓ | **81.4** (+1.3) | **78.1** (+2.1) | **83.8** (+1.5) | 87.6 (-0.7) | 81.3 (-3.5) | **83.9** (+2.8) |
| | EVA-02 ViT-L finetuned | *(6a)* | ✗ | 80.2 (+0.1) | 76.6 (+0.6) | 82.5 (+0.2) | **88.4** (+0.1) | **85.6** (+0.8) | 81.6 (+0.5) |
| | | *(6b)* | ✓ | 80.5 (+0.4) | 77.2 (+1.2) | 83.1 (+0.8) | **88.6** (+0.3) | **86.2** (+1.4) | 82.8 (+1.7) |
| | EVA-02-CLIP ViT-L probed | *(5a)* | ✗ | 81.0 (+0.9) | **78.0** (+2.0) | **83.9** (+1.6) | 87.4 (-0.9) | 79.0 (-5.8) | 82.9 (+1.8) |
| | | *(5b)* | ✓ | 81.2 (+1.1) | **78.2** (+2.2) | **83.7** (+1.4) | 87.4 (-0.9) | 81.7 (-3.1) | **83.5** (+2.4) |
| | EVA-02-CLIP ViT-L finetuned | *(6a)* | ✗ | 79.9 (-0.2) | 76.8 (+0.8) | 82.6 (+0.3) | **88.5** (+0.2) | **85.7** (+0.9) | 81.3 (+0.2) |
| | | *(6b)* | ✓ | 80.5 (+0.4) | 77.0 (+1.0) | 82.9 (+0.6) | **88.8** (+0.5) | **86.4** (+1.6) | 82.7 (+1.6) |

Table H: **Distillation with EVA-02 (Fang et al., 2023) and EVA-02-CLIP (Sun et al., 2023) pretrained models** for classification on DomainNet and fine-grained classification. We report results with and without data augmentation based on Stable Diffusion (SD), using EVA-02 ViT-S as student and EVA-02 ViT-L, EVA-02-CLIP ViT-L as teachers. Relative gains w.r.t. to underlined result from Table G are in parentheses. **Bold** numbers: within 95% confidence interval of the best score for each task.

on datasets composed of 38 million images and 2 billion images respectively, and ii) EVA-02 ViT-S model as a student, pretrained on ImageNet-21k.

Results are reported in Tables G and H. The line numbering is kept consistent with that of DINOv2 experiments. Compared to DINOv2, probing results for ViT-L are stronger on DomainNet but weaker on fine-grained tasks, especially on Aircraft *(2a)*. Probing results for ViT-S are overall very low compared to DINOv2's ViT-S *(1a)*, which is certainly due to the fact that EVA-02's ViT-S was pretrained from scratch while DINOv2's ViT-S was distilled from their ViT-g. Distillation with a probed ViT-L is detrimental for CUB and Aircraft *(5)*. On the four other tasks, it boosts results and outperforms distillation from a finetuned ViT-L *(5 vs 6)*. Similarly to the case of Cityscapes with DINOv2 models (Tables 1 and 2), poor distillation results with the probed ViT-L for CUB and Aircraft can be explained by fact that the finetuned ViT-S student outperforms the probed ViT-L teacher by a large margin (resp. 3.2% and 21.3% accuracy).

### B.4   Ablation for the distillation loss $\mathcal{L}_{\text{distill}}$

We compare the distillation loss originally proposed by Hinton et al. (2015) (and that we used in all experiments in the main paper) to other more recent alternatives from the literature: i) Hard-label distillation, that consists in a cross-entropy loss with the hard label prediction produced by the teacher and is employed by Touvron et al. (2021) for learning their distillation token (here, to be agnostic to the architecture, we simply reuse their loss, not the full distillation protocol), ii) CRD (Tian et al., 2020), that aligns teacher and student representations through a contrastive learning objective, and iii) DKD (Zhao et al., 2022), that decomposes the classical knowledge distillation loss into target-class and non-target-class losses.

|  | $\mathcal{L}_{\text{distill}}$ | Mixup | CUB | Aircraft | DTD |
|---|---|---|---|---|---|
| Finetuning | - | ✓ | 87.3 | 87.8 | 81.6 |
| Distillation | Hard-label KD (Touvron et al., 2021) | ✗ | 88.1 | 88.3 | 81.2 |
|  | CRD (Tian et al., 2020) | ✓ | 89.0 | 88.4 | 83.3 |
|  | DKD (Zhao et al., 2022) | ✗ | 88.5 | 85.7 | 82.8 |
|  | KD (Hinton et al., 2015) | ✗ | 88.6 | 89.0 | 82.6 |
|  |  | ✓ | 89.1 | 89.6 | 83.1 |

Table I: **Choice of distillation loss $\mathcal{L}_{\text{distill}}$.** Comparison of the classical distillation loss KD introduced by Hinton et al. (2015), chosen for our study, with i) hard-label distillation, used by Touvron et al. (2021), ii) CRD (Tian et al., 2020), and iii) DKD (Zhao et al., 2022). We evaluate the version of CRD that uses KD and a temperature $\tau = 0.5$, as it gave the best results. Note that the Mixup augmentation is not applied for hard-label distillation and DKD, as these losses are not compatible with Mixup. Results are with ViT-g as teacher and ViT-S as student, withhout data augmentation based on Stable Diffusion.

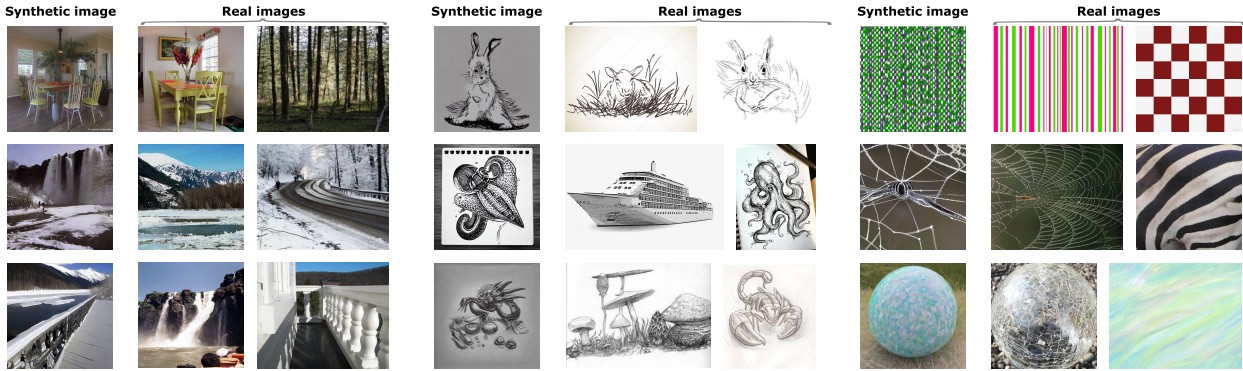

Figure A: **Diffusion-based data augmentation**. Examples of synthetic images generated using ImageMixer (Pinkney, 2022) as described in Section 3.2, mixing two training images from ADE20K (Zhou et al., 2017) (left), Sketch from DomainNet (Peng et al., 2019) (middle) and DTD (Cimpoi et al., 2014) (right). Those populate the extended dataset $\mathcal{D}_{\text{sd}}$ used for distillation.

Results for fine-grained classification are reported in Table I and show that, for the task-specific distillation of DINOv2 pretrained models, the classical knowledge distillation loss introduced by Hinton et al. (2015) performs best.

## C Additional visualizations

Visualizations of the synthetic images produced by our augmentation protocol based on stable-diffusion for ADE2OK (Zhou et al., 2017), DomainNet's Sketch (Peng et al., 2019) and DTD Cimpoi et al. (2014) can be found in Figure A.

