# OpenReview forum: "On Good Practices for Task-Specific Distillation of Large Pretrained Visual Models"
_TMLR — Accepted by TMLR_

### Review · Reviewer_dfB3 · 2024-03-06

**Summary Of Contributions:**

This paper studies the question of distilling the knowledge of large vision models on a specific task to smaller models. Additionally, the paper studies using stable diffusion image generation methods as a data augmentation-like technique. The authors run experiments using the family of VIT models from DINOv2 and a randomly initialized ResNet. They find that linear probing yields a more useful teacher than fine-tuning. Additionally, the paper finds that self-distillation can work remarkably well and that there is no performance gap when distilling directly from very large to much smaller models. Finally, they perform additional analysis and ablations of their results.

**Audience:**

Yes

**Claims And Evidence:**

Yes

**Requested Changes:**

I would like the authors to consider improving the presentation of the main results. For example, having the first column in Table 1 say "Student Arch" rather than just "Arch" would help. Possibly moving some of the info to a different table would help as well -- e.g., self-distillation is an interesting but a bit unintuitive case, and maybe moving all "no-teacher" setups would make the case clearer. It's hard for me to provide more precise feedback without actually changing the paper, but I'd like to highlight that working on the presentation would make it easier to parse for wider audiences.

**Strengths And Weaknesses:**

Overall, I think the paper presents an interesting study of the transfer problem. Although I mention some weaknesses here, I still think it should be interesting to the TMLR’s audience and is technically sound.

Strengths:

- The problem of efficient transfer is very important and solving it has become even more urgent in recent years with the rising popularity of very large models.
- The paper aims to perform a thorough study of the problem and update best practices for researchers and practitioners in the field. They offer several useful takeaways.
- The experiments checking the impact of stable diffusion-based image augmentation are a valuable addition to the analysis. The experiments in Table 2 are especially interesting in this regard.
- To the best of my knowledge, the paper faithfully describes related work.

Weaknesses:

- The presentation of the paper could be improved. It took me some time to learn how to read Table 1 which is the main result of the work.
    - Additionally, I’m not sure how to read Table 4. I assume that the first row ($L_{train}$ only) should correspond to row 3(b) in Table 1, but it clearly doesn’t.
- The paper shows a surprising result that a probed teacher is more useful than a finetuned teacher, but it does not try to explain this phenomenon. In my opinion, the paper already does enough to warrant acceptance in TMLR but explaining this behavior would definitely strengthen the paper.
- The novelty is somewhat limited, but obviously, this is not an important factor in TMLR.

---

> ### Author Response · Authors · 2024-03-26
> **Answer to Reviewer dfB3**
>
> > *The paper shows a surprising result that a probed teacher is more useful than a finetuned teacher, but it does not try to explain this phenomenon.*
>
> We agree with the reviewer that this result is surprising, especially in cases where the finetuned teacher has a much higher accuracy than the probed one.  Following our experiments on DINOv2 and EVA-02, as well as EVA-02-CLIP (added to answer a comment from reviewer FHF3), it seems that this observation holds whenever the probed teacher is not too far behind in accuracy from the finetuned one (less than 6% gap for the experiments we conducted).
>
> This observation could be due to *catastrophic forgetting* [11] when performing finetuning. Indeed, one interpretation could be that, when specializing for a given task, the finetuned teacher has forgotten some features that are not immediately relevant for optimizing the task loss, but that still help generalization.
> For instance, when finetuned on the CUB bird classification task, the pretrained model could end up only relying on *spurious correlations* (such as the background, a standard source of spurious correlations in CUB, as studied in [12]) that provide *shortcuts* to the optimization. These shortcuts help improving the teacher accuracy but are detrimental to the distillation process.
>
> This discussion can be found in an new paragraph *"Probing can yield better teachers than finetuning"* of the Section 5.
>
> [11]: James Kirkpatrick, Razvan Pascanu, Neil Rabinowitz, Joel Veness, Guillaume Desjardins, Andrei A Rusu, Kieran Milan, John Quan, Tiago Ramalho, Agnieszka Grabska-Barwinska, et al. Overcoming catastrophic forgetting in neural networks. Proceedings of the national academy of sciences, 114(13):3521–3526, 2017
>
> [12]: Shiori Sagawa, Pang Wei Koh, Tatsunori B Hashimoto, and Percy Liang. Distributionally robust neural networks for group shifts: On the importance of regularization for worst-case generalization. arXiv preprint arXiv:1911.08731, 2019
>
> > *The presentation of the paper could be improved. It took me some time to learn how to read Table 1 which is the main result of the work*
>
> Thank you for your feedback, we totally agree.
> Following the reviewer’s advice, we have split the former Table 1 into three tables, the new Tables 1, 2 and 3. This allowed splitting ViT experiments between distillation and non-distillation experiments (Tables 2 and 1) and isolating experiments with the ResNet-50 model (Table 3). On top of this new structure of the results, we have changed the table headers to make those more readable. More precisely,
> - Tables 2 and 3 now contain ‘Student’ and ‘Teacher’ columns.
> - The use of data augmentation based on Stable Diffusion is indicated in a separate column in Tables 2 and 3.
> - For each teacher model in Tables 2 and 3, we indicate whether is was probed or finetuned.
>
> We updated the text and all other tables according to this new formatting.
>
> > *I’m not sure how to read Table 4. I assume that the first row (L_train only)  should correspond to row 3(b) in Table 1, but it clearly doesn’t.*
>
> We thank the reviewer for their careful review of the paper. We made a mistake when reporting finetuning results for Tables 3 and 4 (now Table 5 and 6); they should indeed correspond to line *3b* (now *1b*) in Table 1. This is fixed. We have double-checked the remaining lines to ensure they are correct.

---

### Review · Reviewer_QYsY · 2024-03-08

**Summary Of Contributions:**

The paper studies “task-specific distillation” - distilling a (typically large) teacher model to a student model on a specific task. The teachers are pre-trained on DINOv2 and the downstream tasks are several classification datasets and several semantic segmentation datasets. The paper studies transfer from teachers of different sizes to students of different sizes in different settings and additionally proposes a data augmentation stratedg based on using a diffusion model for mixing real images. The paper shows that the proposed augmentation strategy is helpful and come to several conclusions about task-specific distillation in the considered setting.

**Audience:**

Yes

**Broader Impact Concerns:**

No specific concerns

**Claims And Evidence:**

Yes

**Requested Changes:**

1. Compare to relevant baselines and/or state of the art (Con 2)
2. Very desirably, run at least some experiments with CLIP-style models (Con 3) - do not have to be as thorough as the current DINOv2 ones, but some
3. Address the "more minor cons", mostly those are presentation issues

**Strengths And Weaknesses:**

Pros:
1. Well written, easy to read
2. Thorough experiments, evaluation on many datasets, appropriate comparisons, and ablation studies
3. Reasonable gains from the proposed techniques - the data augmentation strategy and distilling a “probe-tuned” teacher instead of a fully fine-tuned one

Cons:
1. The gains compared to a fine-tuning baseline are not huge (for pre-trained ViT-S, they are somewhat larger for the from-scratch models)
2. On a related topic, I am really missing some baselines. While the study in the paper can be seen as self-contained and interesting of itself, and it does not necessarily need to beat state of the art, I find that its relevance is difficult to judge without knowing how the reported results compare to some relevant baselines in the field (including SOTA) - and I urge the authors to include some of those. Not proposing specific papers because I am not familiar enough with this sub-field, but I am sure the authors can find relevant baselines to report.
3. It would be really helpful to report results with multiple types of teachers, not only DINOv2. Are the proposed techniques still relevant to e.g. CLIP-style models?


More minor cons:

4. Some findings are formulated in an overly general way, not justified by the evidence. For instance, “Linear probing generally yields better teachers than finetuning” is only shown on DINOv2 and should not be stated like this (it is somewhat clarified in the text, but still, shouldn’t be stated so generally in the first place)
5. A few times the term “stable diffusion” is used in the context where, it seems, just any diffusion model is meant - it’s unclear why “stable” has to be there. If referring to just some diffusion model, please just write “diffusion”, and if referring to specifically Stable Diffusion, please write so, with capitalization.
6. In Figure 2, seems both the augmented and the non-augmented image go into the student, what does that mean? The text seems to suggest that it’s the augmented image in both cases “Note that we feed the same augmented images to both the teacher and the student models”
7. “Linear probing runs for 20 epochs” → it’s not actually linear, but MLP?
8. How were the durations (#epochs) of probing and fine-tuning selected? They can play a huge role for the final results

---

> ### Author Response · Authors · 2024-03-26
> **Answer to Reviewer QYsY (1/2)**
>
> > *[Weakness 1] The gains compared to a fine-tuning baseline are not huge for pretrained ViT-S*.
>
> We agree with this observation. However, please note that DINOv2’s ViT-S model is the strongest ViT-S model to date, meaning that finetuning ViT-S already gives an excellent baseline, yet task-specific distillation allows pushing results even further.
>
> > *[Weakness 2]  I am really missing some baselines [...] I find that [the paper’s] relevance is difficult to judge without knowing how the reported results compare to some relevant baselines in the field (including SOTA) - and I urge the authors to include some of those.*
>
> Thank you for the suggestion. First, note that the paper already includes a number of strong baselines: lines *1a*, *1b*, and *6a* (see details below). Second, we have added several comparisons with more recent distillation strategies and showed that the one we use for our experimental protocol performs best.
>
> More precisely, our study starts from the pretrained models provided by the authors of DINOv2 and EVA-02. Note that DINOv2's ViT-S and ViT-L models were distilled from their ViT-g using a simple output-based distillation. Yet DINOv2's ViT-S model is the best publicly available today, making the finetuned ViT-S an excellent baseline.
>
> Our protocol performs task-specific distillation with teacher probing, using the classical distillation loss introduced by Hinton et al. [1]. This loss is still broadly used today, including in approaches similar to ours, see for instance [5][6].  We have compared Hinton et al.’s loss with three alternatives: i) the hard-distillation loss used in [DeiT](https://arxiv.org/abs/2012.12877) [2] to learn their distillation token, ii) [Contrastive Representation Distillations (CRD)](https://arxiv.org/abs/1910.10699) [3], and iii) [Decoupled Knowledge Distillation (DKD)](https://arxiv.org/abs/2203.08679) [4]. These additional experiments can be found in Section B.4 of the appendix, and show that the strategy we use performs best in the context of our study.
>
> Alternative distillation approaches that are already part of our reported results and could be considered as baselines include:
> - Task-agnostic distillation, followed by student probing or finetuning. This corresponds to lines *1a* and *1b* of Table 1, and this is the distillation strategy followed by DINOv2 as well as by e.g. [5][6][7]
> - Task-agnostic distillation followed by task-specific distillation using a finetuned teacher. This corresponds to line *6a* of Table 2, and this is the distillation strategy followed by e.g. [8][9][10].
>
> [1]: Geoffrey Hinton, Oriol Vinyals, and Jeff Dean. Distilling the knowledge in a neural network. arXiv preprint arXiv:1503.02531, 2015
>
> [2]: Hugo Touvron, Matthieu Cord, Matthijs Douze, Francisco Massa, Alexandre Sablayrolles, and Herve Jegou. Training data-efficient image transformers & distillation through attention. In Proceedings of the International Conference on Machine Learning (ICML), 2021
>
> [3]: Yonglong Tian, Dilip Krishnan, and Phillip Isola. Contrastive representation distillation. In Proceedings of the International Conference on Learning Representations (ICLR), 2020
>
> [4]: Borui Zhao, Quan Cui, Renjie Song, Yiyu Qiu, and Jiajun Liang. Decoupled knowledge distillation. In Proceedings of the IEEE/CVF Conference on Computer Vision and Pattern Recognition (CVPR), 2022
>
> [5]: Siqi Sun, Yu Cheng, Zhe Gan, and Jingjing Liu. Patient knowledge distillation for BERT model compression. In Proceedings of the Conference on Empirical Methods in Natural Language Processing (EMNLP), 2019
>
> [6]: Kan Wu, Jinnian Zhang, Houwen Peng, Mengchen Liu, Bin Xiao, Jianlong Fu, and Lu Yuan. TinyViT: Fast pretraining distillation for small vision transformers. In Proceedings of the European Conference on Computer Vision (ECCV), 2022
>
> [7]: SEED: Self-supervised distillation for visual representation. In Proceedings of the International Conference on Learning Representations (ICLR), 2021
>
> [8]: Victor Sanh, Lysandre Debut, Julien Chaumond, and Thomas Wolf. 2019. DistilBERT, a distilled version of BERT: Smaller, faster, cheaper and lighter. Retrieved from https://arXiv:1910.01108
>
> [9]: Xiaoqi Jiao, Yichun Yin, Lifeng Shang, Xin Jiang, Xiao Chen, Linlin Li, Fang Wang, and Qun Liu. Tiny-BERT: Distilling BERT for natural language understanding. In Findings of the Association for Computational Linguistics: EMNLP, 2020
>
> [10]: Wei Huang, Zhiliang Peng, Li Dong, Furu Wei, Jianbin Jiao, and Qixiang Ye. Generic-to-specific distillation of masked autoencoders. In Proceedings of the IEEE/CVF Conference on Computer Vision and Pattern Recognition (CVPR), 2023

---

> ### Author Response · Authors · 2024-03-26
> **Answer to Reviewer QYsY (2/2)**
>
> > *[Weakness 3] It would be really helpful to report results with multiple types of teachers, not only DINOv2. Are the proposed techniques still relevant to e.g. CLIP-style models?*
>
> Thank you for this great suggestion. We added new experiments in the appendix using the CLIP-pretrained version of EVA-02 models (Section B.3, Tables G,H). These new results complement the experiments already conducted with DINOv2 (in the main text) and EVA-02 (MIM-pretrained models, reported in the appendix).
>
> Our experiments show that on tasks where the accuracy of a probed model is not too far behind that of a finetuned model (less than 6% gap for the experiments we conducted), the former makes a better teacher than the latter. This observation holds both for DINov2 and EVA-02 experiments. For DINOv2, the only case where finetuning makes a better teacher than probing is that of Cityscapes, where the finetuned ViT-L is 8.2% higher in mIoU than the probed ViT-L.
>
> > *[Weakness 4] Some findings are formulated in an overly general way, not justified by the evidence. For instance, “Linear probing generally yields better teachers than finetuning” is only shown on DINOv2 and should not be stated like this.*
>
> We clarified the setting in which our findings hold. In particular we updated the title, and rewrote part of the introduction (to answer this comment, and a similar comment from  Reviewer FHF3). We clarified that our study is specific to computer vision models with generalization capabilities comparable to that of DINOv2 or EVA-02 and have softened the first claim accordingly.
>
> > *[Weakness 5] If referring to specifically Stable Diffusion, please write so, with capitalization.*
>
> Thanks for spotting this. We have corrected it in the whole manuscript.
>
> > *[Weakness 6] In Figure 2, seems both the augmented and the non-augmented image go into the student, what does that mean? The text seems to suggest that it’s the augmented image in both cases “Note that we feed the same augmented images to both the teacher and the student models”*
>
> Thank you for your feedback. We acknowledge that Figure 2 and its caption were ambiguous, they have been revised accordingly.
>
> Note that both original images and the synthetic ones obtained with Stable Diffusion still undergo a classical data augmentation step (such as RandomResizedCrop or ColorJitter). The same standard data augmentations are applied to both the student and the teacher models, *i.e.* both models receive exactly the same batch of augmented images as input. This standard data augmentation is not represented in Figure 2 but now mentioned in the caption.
>
> In Figure 2, the two different arrows the student receives are to distinguish the data used to optimize the task loss (only original images, shown in blue) and the data used to optimize the distillation loss (both original and synthetic images, *i.e.* the extended set, shown in orange). We clarified this by slightly reworking the figure and the caption.
>
> > *[Weakness 7] “Linear probing runs for 20 epochs” → it’s not actually linear, but MLP?*
>
> Thanks for this great catch. The probing is only linear for segmentation, an MLP is used for all classification tasks. We changed the text to the more general ‘probing’ term, including in the first claim of our study.
>
> > *[Weakness 8] How were the durations (#epochs) of probing and fine-tuning selected?*
>
> The number of epochs was chosen over a subset of tasks as follows:
> - For the student model, we chose 30 epochs for probing and 80 epochs for finetuning based on validation accuracy. We ran hyperparameter tuning (of learning rate and weight decay) for each choice of #epochs, and selected the number of epochs that reached the highest validation accuracy overall.
> - For the teacher model, we chose 20 epochs for probing and 50 epochs for finetuning based on the student’s validation accuracy after distillation, with the same protocol as above.
>
> Although these values may not be optimal for each task individually, we found them to be a good tradeoff over all tasks. We favored using fixed values for the number of epochs, and only cross-validated the weight decay and learning rate hyperparameters, as is traditionally done in the literature.

---

### Review · Reviewer_FHF3 · 2024-03-12

**Summary Of Contributions:**

The paper offers a systematic overview of techniques that can used to learn a smaller models from a large pretrained models (mostly knowledge distillation, also using fine-tuning and data augmentation). Across their tasks, the paper shows several findings that achieve universal performances for the datasets and settings they have tried. The paper conclusion regarding how to train smaller models in general seems not supported.

**Audience:**

Yes

**Broader Impact Concerns:**

none noted.

**Claims And Evidence:**

No

**Requested Changes:**

1. I would recommend the author to rephrase this into more like a method paper, focusing particular applications and phrasing the study into a new method, which might be OK as a novel contribution, instead of a systematic study, so that the paper will be free of the issues of overclaiming.
   - if the author prefer the current wording, the work might need substantially more experiments to live to its claims.

2. the paper will need thorough studies of how much the stable diffussion contributes to the learning, with or without distillation. In particular, answering the question that stable diffusion model likely has seen more data than teacher model, why do we still need the teacher model.

**Strengths And Weaknesses:**

Strengths:
1. the paper evaluates a big topic regarding how to learn smaller models given a pretrained model, and it covers several topics regarding knowledge distillation, data augmenation, and fine-tuning.

2. the paper offers several practical guidelines that might benefit the community of similiar interests.

Weakness

1. the claim of the paper seems a bit way too big. While the paper indeed evaluates a couple of datasets, from image classification to semantic segmentation, the community have studied knowledge distillation way bigger than this scope. Although the paper can achieve consistent findings in these topics, it's not convincing enough the paper's conclusion can live upto its claim.
     - tons of backbone choices, tasks choices, and other minor changes of the method might affect the effectiveness of the method. In addition, several parts of the paper is written in a fairly generic way, while the contents are mostly for computer vision studies.

2. the data augmentation with stable diffusion is fairly strong. It is reasonable to believe that data augmentation with stable diffusion itself can benefit strongly to the improvements of the model, for the reason that stable diffusion model has also seen tons of data, probably even more than the teacher model per se.

---

> ### Author Response · Authors · 2024-03-26
> **Answer to Reviewer FHF3**
>
> > *[Weakness 1] The claim of the paper seems a bit way too big. [...] Tons of backbone choices, tasks choices, and other minor changes of the method might affect the effectiveness of the method. In addition, several parts of the paper is written in a fairly generic way, while the contents are mostly for computer vision studies.*
>
> Indeed, the manuscript is restricted to the computer vision domain, and we apologize if this was not stated clearly enough. To avoid any misunderstanding, we have adjusted the title and rewritten part of the introduction to clarify the scope of our claims. In the introduction, we have also emphasized the fact that our study is conducted on the most common backbones (ViT-S to ViT-g and ResNet-50) and on several large models with good generalization capabilities such as DINOv2 (in the main paper), but also EVA-02 and EVA-02-CLIP, that we have added to answer a comment from QYsY (in the appendix). We have also adjusted  the first claim to account for a similar comment from Reviewer QYsY.
>
> > *[Weakness 2] The paper will need thorough studies of how much the stable diffusion contributes to the learning, with or without distillation. In particular, answering the question that stable diffusion model likely has seen more data than teacher model, why do we still need the teacher model.*
>
> Thank you for this comment. To clarify the complementary role of augmentations based on Stable Diffusion and teacher-student distillation, we have added a separate paragraph in the discussion section (Section 5, paragraph *Stable Diffusion as a source of additional information*) that extends the existing discussion in the related work (paragraph *Data augmentation with Stable Diffusion*).
>
> More precisely, the experiments that we report in Table 5 suggest that without a teacher, it may not be possible to improve results by using our prompt-free augmentation method based on Stable Diffusion. Indeed, Table 5 shows that this augmentation strategy is only beneficial in the context of distillation: using it for optimizing the task loss, with or without the teacher, degrades performance.
>
> Finally, note that all our distillation experiments are conducted with and without data augmentation based on Stable Diffusion, which allows us to precisely measure the gains that are specific to this augmentation strategy.

---

### Author Response · Authors · 2024-03-26
**Common answer**

Dear Editor and Reviewers,

First, we would like to thank all reviewers for their insightful and constructive comments that helped improve our work.

We have carefully taken into account all suggestions and have uploaded a revised version of our manuscript. All the modifications are colored in blue to allow for an easier review.
Among the key modifications, we have:
- *Clarified the scope*: we have modified the title and the introduction to better contextualize our study
- *Consolidated the experimental validation*: we have performed additional experiments with a CLIP teacher (EVA-02-CLIP) and have added comparisons with other more recent distillation losses.
- *Improved the presentation of our results*: we have replaced the large Table 1 with several shorter tables that are easier to parse, and have adapted the discussions and all other tables accordingly.
- *Extended the discussion section*, with two additional paragraphs: "*Probing can yield better teachers than finetuning*", and "*Stable Diffusion as a source of additional information*".

We answer to each reviewer individually below.

---

### Decision · Action_Editor_go1x · 2024-04-15

**Recommendation:** Accept as is

**Comment:**

As described above, the reviewers were mostly convinced after the discussion phase that the paper provides interesting claims with sufficient evidence.

**Audience:**

All reviewers agree that the topic is very timely and interesting for the TMLR audience.

**Claims And Evidence:**

The reviewers were initially skeptical of the scope of the paper and have voiced concerns that the authors might have tried to put too much content into a single submission, but after a very active rebuttal phase, most reviewers are now convinced that the scope has been narrowed down appropriately. Moreover, the reviewers commend the thorough experiments and solid technical work in the paper.